# Disentangling Intrinsic Importance from Emergent Structure in Multi-Expert Orchestration

**Sudipto Ghosh**[*]                                                    *sudipto.ghosh@scai.iitd.ac.in*
*Yardi School of Artificial Intelligence*
*Indian Institute of Technology Delhi, India*

**Sujoy Nath**[*]                                                       *sujoynathofficial@gmail.com*
*Department of Electrical Engineering*
*Indian Institute of Technology Delhi, India*

**Sunny Manchanda**                                                    *sunny.dysl-ai@gov.in*
*DRDO Young Scientist Laboratory – Artificial Intelligence*
*Defence Research and Development Organisation, India*

**Tanmoy Chakraborty**                                                 *tanchak@iitd.ac.in*
*Department of Electrical Engineering*
*Yardi School of Artificial Intelligence*
*Indian Institute of Technology Delhi, India*

**Reviewed on OpenReview:** *https://openreview.net/forum?id=4W7sgat04A*

## Abstract

Multi-expert systems, where multiple Large Language Models (LLMs) collaborate to solve complex tasks, are increasingly adopted for high-performance reasoning and generation. However, the orchestration policies governing expert interaction and sequencing remain largely opaque. We introduce INFORM, an interpretability analysis that treats orchestration as an explicit, analyzable computation, enabling the decoupling of expert interaction structure, execution order, and functional attribution. We use INFORM to evaluate an orchestrator on GSM8K, HumanEval, and MMLU using a homogeneous consortium of ten instruction-tuned experts drawn from LLaMA-3.1 8B, Qwen3 8B, and DeepSeek-R1 8B, with controlled decoding-temperature variation, and a secondary heterogeneous consortium spanning 1B–7B parameter models. Across tasks, routing dominance is a poor proxy for functional necessity. We reveal a divergence between *relational importance*, captured by routing mass and interaction topology, and *intrinsic importance*, measured via gradient sensitivity: frequently selected experts often act as interaction hubs with limited influence, while sparsely routed experts can be structurally critical. Orchestration behaviors emerge asynchronously, with expert centralization preceding stable routing confidence and expert ordering remaining non-deterministic. Targeted ablations show that masking intrinsically important experts induces disproportionate collapse in interaction structure compared to masking frequent peers, confirming that INFORM exposes functional and structural dependencies beyond accuracy metrics alone.

## 1 Introduction and Prior Art

Large Language Models (LLMs) are increasingly deployed not as standalone solvers but as components within multi-expert and multi-agent systems, where multiple models interact to solve complex reasoning tasks (Wu et al., 2024; Hong et al., 2024; Qian et al., 2024). Rather than relying on a single monolithic model, these

---

[*]Equal contribution.

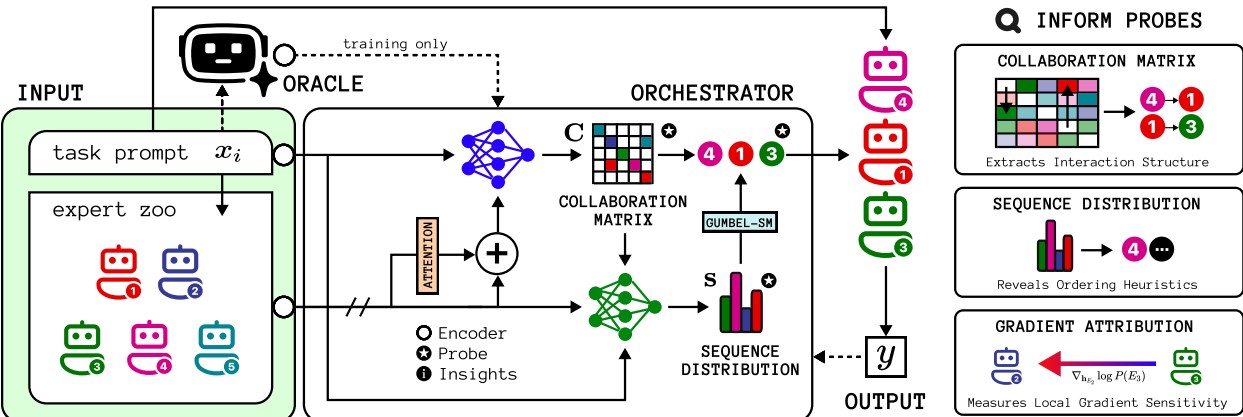

Figure 1: **Probing multi-expert orchestration with INFORM**. The figure illustrates the insights extracted during inference: (1) Probing the interaction module reveals the collaboration topology; (2) Analyzing the selection mechanism exposes ordering heuristics; and (3) Backpropagating decisions to expert representations isolates intrinsic attribution distinct from observed routing frequency. The symbol `//` denotes a collection of outputs produced by multiple experts.

systems coordinate experts through an orchestration mechanism (Dang et al., 2025) that determines which expert is invoked, in what order, and under what context. This paradigm has enabled strong empirical gains across reasoning, coding, and decision-making benchmarks (Lei et al., 2024; Islam et al., 2024). Existing approaches to orchestration span several design philosophies. Some systems rely on externally controlled execution graphs that explicitly manage state and control flow (Wu et al., 2024; LangChain AI, 2025), while others encode coordination through role-based prompting or social simulation (Li et al., 2023; Qian et al., 2024). More recent work introduces learned or routing mechanisms that dynamically select experts during inference (Chen et al., 2024; Ong et al., 2025; Wang et al., 2025). Despite their architectural differences, these methods share a common assumption: orchestration policies are optimized for performance but are rarely examined as objects of analysis.

**Related Work.** Conditional computation has long been studied to improve efficiency and specialization, exemplified by the Mixture-of-Experts (MoE) paradigm, which routes inputs to subsets of parameters (Shazeer et al., 2017; Fedus et al., 2022) or selects among multiple frozen language models (Chen et al., 2024; Ong et al., 2025). While systems like LLM-Blender (Jiang et al., 2023b) focus on aggregating outputs, existing routing frameworks often optimize implicitly through task loss, offering limited visibility into the functional contribution of experts or the temporal dependencies of routing decisions. This issue of opaque orchestration becomes even more pronounced in multi-agent collaboration, where systems range from rigid, centralized workflows to decentralized networks (Tran et al., 2025; Guo et al., 2024; Hong et al., 2024) that are vulnerable to error propagation from faulty agents (Huang et al., 2025). In contrast to approaches that rely on static aggregation or rigid protocols, our analysis emphasizes that orchestration is inherently sequential, and INFORM specifically focuses on diagnosing expert influence within these interactions without modifying the collaboration protocol itself. Appendix B provides extended related work.

**Motivation and Crisis.** The internal logic of orchestration remains largely opaque in current frameworks. It is often unclear why certain experts are selected, whether expert ordering reflects genuine functional dependencies or incidental heuristics, and to what extent routing frequency corresponds to necessity. Many approaches abstract routing as order-invariant (Jiang et al., 2023b; Chen et al., 2024), despite orchestration being inherently sequential (Yao et al., 2025). This opacity makes it difficult to distinguish specialization from redundancy, and to diagnose failure modes such as brittle routing, degeneration into static ensembles (Vallecillos-Ruiz et al., 2025), or silent cost inflation (Han et al., 2024; Xi et al., 2025). This lack of interpretability is a practical limitation as orchestration decisions directly affect reliability, efficiency, and safety, particularly in high-stakes or tool-augmented settings (Bommasani et al., 2021). Understanding orchestra-

tion requires tools that expose expert interactions and sequencing rather than treating routing as a black box (Rudin, 2019).

**Our Contributions.** In this work, we introduce INFORM, an interpretability analysis attempt at analyzing neural orchestration mechanisms in multi-expert LLM systems. INFORM treats orchestration as an explicit, analyzable computation, enabling systematic inspection of how experts are selected, how they interact over time, and how sequencing patterns emerge. Using INFORM, we seek to answer the following questions:

**RQ1:** What structured interaction and specialization patterns emerge in neural orchestration policies during training, and how do these patterns differ across tasks?

**RQ2:** How does the orchestrator learn and employ expert sequencing decisions, and are ordering preferences stable, adaptive, or non-deterministic at inference time?

**RQ3:** Does observed routing behavior reflect intrinsic expert importance, or do routing frequency and intrinsic attribution diverge in practice?

**RQ4:** How robust and functionally grounded are orchestration decisions under targeted perturbations and expert-level interventions?

Table 1 shows a comparison between INFORM and other frameworks. We analyze a learned orchestrator coordinating ten frozen, instruction-tuned LLM experts. For our experiments, we use a homogeneous pool of models with similar parameter counts. We use the term *homogeneous* to denote parity in model parameter capacity, rather than behavioral uniformity, as functional diversity is maintained through controlled decoding-temperature variation. To validate the generalizability of our findings, we also evaluate a heterogeneous consortium comprising experts with varying parameter counts. Evaluating the orchestration with INFORM on MMLU, HumanEval and GSM8K, reveals consistent but task-dependent orchestration patterns. Routing confidence and specialization emerge gradually, while expert ordering remains structured but non-deterministic. Frequently selected experts are not always intrinsically important, and alignment between intrinsic attribution and routing varies substantially across tasks. Our analysis yields three key insights. First, we reveal a divergence between *relational importance* and *intrinsic importance*: orchestrators often utilize interaction hubs – experts routed to frequently but with limited functional influence. Second, orchestration behaviors emerge asynchronously, establishing *who* to trust before resolving *how confidently* to route. Finally, perturbation studies confirm INFORM captures genuine structural necessity; masking the most intrinsically important expert on MMLU induces $5.5\times$ higher routing KL divergence than sequencing divergence, validating these experts as critical structural dependencies. Our gradient-based approach captures local sensitivity of routing decisions rather than full causal structures. It provides functional attribution to diagnose structural dependencies without claiming formal guarantees. We address commonly asked questions and present key takeaways in Appendix A. Statistical tests supporting the claims are reported in Appendix J. Appendix K discusses INFORM's applicability to black-box and API-based environments. Appendix L details the individual baseline performance scores for all experts in the homogeneous consortium. Appendix M validates the robustness of our intrinsic-importance metric across alternative encoders and pooling strategies. Finally, Appendix N provides the exact prompts utilized for comparison with MetaGPT.

## 2   INFORM: An Interpretable Framework for Orchestrating Multi-Expert Systems

We introduce INFORM, an interpretability analysis designed to peek inside an orchestrator for multi-expert systems. To demonstrate the capabilities of INFORM, we apply it to a representative orchestration setup as outlined in Figure 1. When an user submits a prompt, an effective orchestrator routes this prompt through a sequence of specialized experts. INFORM extracts insights at three key stages of this inference process: (i) to observe how experts pass information (*e.g.*, $E_1$ handing off to $E_2$), (ii) to observe the selection of the initial expert (*e.g.*, recognizing that $E_1$ must take the first shot at this task), and to identify dependencies between routing decisions and expert representations (*e.g.*, assessing whether routing decisions are sensitive to $E_2$'s output when selecting $E_3$, rather than just guessing). In this section, we describe the canonical orchestrator

Table 1: **Landscape of routing and coordination methods with respect to orchestration transparency.** Most existing systems provide limited or indirect insight into coordination behavior, focusing primarily on cost, performance, or engineering convenience. The reference orchestration setup for `INFORM` stands apart by explicitly allowing gradient-based functional attribution and structural interpretability of expert interactions.

| Method | Primary Focus | Coordination Type | Interpretability Emphasis |
|---|---|---|---|
| LLM-Debate (Estornell & Liu, 2024) | Multi-agent debate paradigm | Agents generate and critique to converge on responses | Low – debates do not expose internal decisions |
| Mixture-of-Experts (Shazeer et al., 2017) | Distributed expert selection within models | Expert token routing within MoE layers | Moderate – some analysis of routing behavior in models |
| RouteLLM (Ong et al., 2025) | Routing for cost/performance tradeoff | Router selects between stronger/weaker LLMs | Moderate – routing decisions based on preference data can be evaluated |
| IRT-Router (Song et al., 2025) | Interpretable LLM router | Trains router with Item Response Theory | High – explicit interpretable ability/difficulty metrics |
| MetaGPT (Hong et al., 2024) | Multi-agent collaboration using SOPs | Structured agent workflows with predefined roles | Very Low – minimal formal interpretability |
| AutoGen (Wu et al., 2024) | Multi-agent AI workflows | Conversational agent orchestration with message passing | Low – primarily engineering ease of building agent dialogue workflows |
| FrugalGPT (Chen et al., 2024) | Cost-efficient cascade of LLMs | Sequential cascade routing until satisfactory response | Low – focuses on cost/performance rather than deep analysis |
| DyLAN (Liu et al., 2023) | Dynamic LLM-agent network | Task-adapted agent selection and interaction | Low – focuses on performance/efficiency improvements |
| Our Setup | Interpretability of orchestration logic | Explicit analyzable orchestration with interaction | Very High – attribution-based, interaction structure, sequencing |

$\mathcal{O}_\theta$, and subsequently detail our methodology used to probe the orchestrator, separating learned interaction structures from intrinsic attribution. [2]

## 2.1 Orchestration Setup

We design an orchestrator $\mathcal{O}_\theta$ that manages an expert consortium $\mathcal{E} = \{E_1, \ldots, E_N\}$, which represents a generic class of differentiable routers that optimize sequential execution. Each expert is an autoregressive language model.

**Model.** The orchestrator $\mathcal{O}_\theta$ operates on latent representations of the input prompt $x$ and initial tokens generated by the experts. It utilizes a frozen BERT-based (Devlin et al., 2019) encoder to project expert outputs onto a common representation space. `INFORM` provides a toolkit to probe $\mathcal{O}_\theta$ by extracting and analyzing the intermediate signals produced during inference. The orchestration logic is implemented using two modules:

1. **Interaction Module.** To disentangle general reasoning from coordination logic, we process expert representations **h** through a *Routing Adapter*, which projects them to a routing space. We compute a conditional transition matrix $\mathbf{C}(x) \in [0,1]^{N \times N}$ using query-key attention, augmented by a static semantic prior to stabilize early training:

$$\mathbf{C}_{ij}(x) \propto \text{Softmax}\left(\frac{Q(h_i)K(h_j)^T}{\sqrt{d}} + \lambda \cdot \text{CosSim}(\mathbf{h})\right)$$

where $h_i$ and $h_j$ are the encoded representations of expert $E_i$'s and $E_j$'s output in the routing space, CosSim represents the cosine similarity, and $\lambda$ is a learnable scalar that scales the influence of semantic similarity on the collaboration topology. Softmax is applied row-wise over $j$ for each source expert $E_i$. This matrix captures the learned compatibility between $E_i$ and $E_j$.

---

[2]Source Code is available at https://github.com/parmanu-lcs2/inform.

2. **Selection Module.** A separate prediction head computes the marginal selection distribution $\mathbf{s}(x) \in [0,1]^N$ using global connectivity of each expert derived from $\mathbf{C}(x)$. This component determines the optimal experts to invoke given the current context using the Gumbel-Softmax trick, allowing for differentiable sampling during training. The probability of selecting expert $i$ is given by:

$$P(E_i|x) = \text{GumbelSoftmax}\left(h^{(i)} + \phi^{(i)} + \frac{1}{2}\sum_j(\mathbf{C}_{ij} + \mathbf{C}_{ji}) - \gamma \cdot t\right)$$

where $h^{(i)}$ is the projection of the encoded input, $\phi^{(i)}$ is a learned scalar that estimates the quality of the selection, $t$ is the position of the expert $E_i$ in the sequence, and $\gamma$ is a penalty multiplier to discourage longer sequences.

**Training.** The orchestrator is trained using a composite objective that maximizes task performance along with alignment of the final system output, to a much larger oracle LLM, which is completely absent during inference. Symmetry enforcement and sparsity penalties are also employed to mimic the efficiency constraints typical of real-world deployments. The details of the orchestrator training are provided in Appendix C. All optimization settings and hyperparameters are summarized in Table 4. We analyze this orchestrator in this work, however, INFORM explicitly does not require attention-based interaction, or oracle distillation; any differentiable routing policy shall suffice.

## 2.2 Method

**Probing Expert Interaction Structure.** We interpret the transition matrix $\mathbf{C}(x)$ as a directed, weighted graph over experts. INFORM analyzes the entropy and rank of this matrix to determine the rigidity of the collaboration structure. For each expert $E_j$, we compute the *Relational Importance* as the total incoming mass $u_j(x) = \sum_{i \neq j} \mathbf{C}_{ij}(x)$. This reveals whether an expert functions as a universal successor, in which case the incoming mass from diverse sources to that expert will be high, or a specialist in isolation.

**Probing Sequencing Decisions.** To understand ordering dynamics, INFORM isolates the marginal selection distribution $\mathbf{s}(x)$. As the initial expert conditions the intermediate representation for the entire chain, we analyze the entropy of $\mathbf{s}(x)$ to detect the emergence of initializers from the expert pool. INFORM disentangles whether an expert is preferred because it is globally competent or because it is actually necessary in the given context.

**Functional Attribution via Gradient Sensitivity.** INFORM decouples *observed usage* from *functional necessity*. We compute *Intrinsic Expert Importance* by backpropagating the selection decisions to the expert representations. We calculate the gradient norm of the log-probability of the selected expert with respect to the representation $h_i$:

$$\mathcal{I}(E_i) = \|\nabla_{h_i} \log P(E_i \mid x)\|_2$$

Gradients are computed with respect to the selection logits for the selected expert at each step and averaged across steps and inputs. This gradient-based attribution measures the degree to which the semantic content of an expert influences the orchestrator's decision. It differentiates between experts selected due to content dependence and those selected using heuristics. By comparing *Relational Importance* with *Intrinsic Importance*, we identify alignment gaps where the orchestrator directs to specialists upon whom it does not fundamentally depend, indicating inefficiency or model hallucination.

# 3 Foundation Behind Our Proposed Design

To justify the need for interpretable probing mechanisms, we first establish that the underlying collaboration dynamics contain discernible, non-trivial patterns. This section presents preliminary studies that serve as the empirical foundation for our proposed design. By characterizing **(i) the structure of expert interactions**, and **(ii) the sensitivity of inference-time sequencing**, we demonstrate that interpretable architectures are not merely a tool for transparency, but a necessity for debugging and optimizing collaborative pipelines.

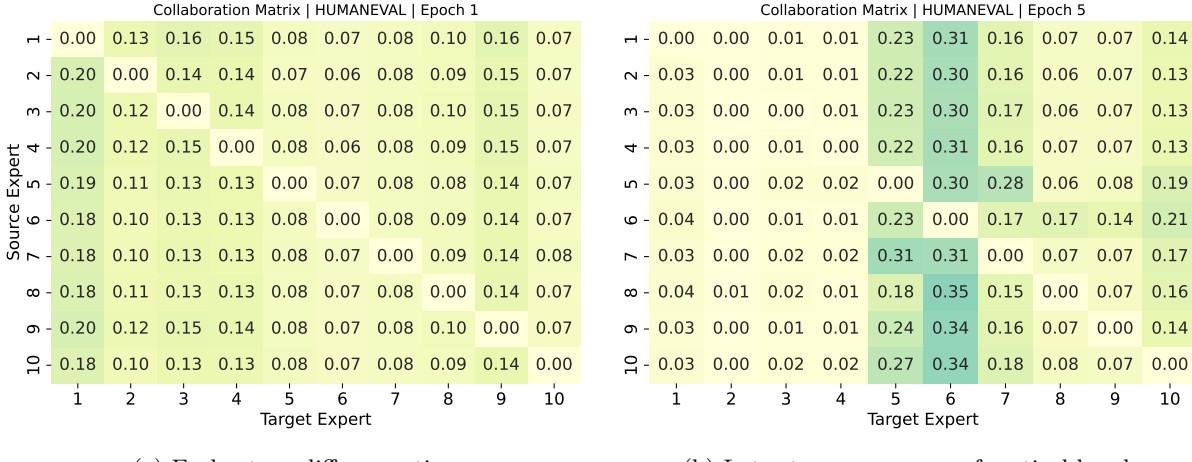

(a) Early-stage diffuse routing
(b) Late-stage emergence of vertical bands

Figure 2: **Evolution of the collaboration matrix $\mathbf{C}(x)$ during training, averaged over the test set.** (a) At the start of training, the orchestrator explores random connections, resulting in a diffuse matrix. (b) By Epoch 5, vertical bands appear which indicates that the orchestrator has identified universal successors.

**Study I: Identifying Redundant Experts via Collaboration Matrices.** We examine the evolution of the orchestrator's collaboration matrices during training on HumanEval to assess whether all experts contribute meaningfully to performance. Early in training, routing is diffuse, reflecting exploratory expert transitions. As training progresses, the matrix becomes highly concentrated, with prominent vertical bands indicating that a small subset of experts consistently receives most routing probability regardless of the source expert (Figure 2). These experts act as universally effective successors for HumanEval. In contrast, several experts receive negligible probability mass, suggesting limited marginal utility in later inference stages. Overall, the orchestrator implicitly identifies a reduced effective expert set. The collaboration matrix, therefore, provides an interpretable signal of functional redundancy, motivating probe-able orchestration mechanisms for principled expert pruning and more efficient inference.

**Study II: Learning to Sequence Experts.** Beyond expert selection, invocation order is critical in multi-stage inference. To assess whether the orchestrator learns meaningful sequencing preferences, we analyze the evolution of the sequence distribution $\mathbf{s}(x)$ across training epochs for HumanEval (Table 2). Early in training, the starting-expert distribution is diffuse, reflecting exploration; by the end, probability mass concentrates on a small set of experts. This shift indicates that the orchestrator differentiates experts not only by overall utility, but by their suitability as *initializers*. In sequential inference, the first expert plays a disproportionate role, as its output conditions all subsequent generations. A weak initializer can introduce errors or biases that downstream experts cannot fully correct, whereas a strong initializer stabilizes the inference trajectory. While the emergence of a dominant initializer shows sensitivity to ordering, the underlying cause remains opaque. These decisions must be attributed to identifiable properties, such as error patterns or representational alignment, rather than treated as black-box behavior.

## 4 Experiments

**Experimental Setup.** Our expert zoo consists of multiple instances drawn from three instruction-tuned model families – LLaMA 3.1 8B (Grattafiori et al., 2024), Qwen3 8B (Yang et al., 2025a), and DeepSeek-R1 8B (Guo et al., 2025), with controlled variation in decoding temperature as listed in Table 6 (Appendix D). This *homogeneous consortium* results in a set of experts that are behaviorally distinct due to stochastic decoding yet possess similar capacity (∼8B), allowing us to study whether the orchestrator can distinguish among experts based on subtle functional differences rather than distinct capability gaps. All experiments are conducted on MMLU (Hendrycks et al., 2021), HumanEval (Chen et al., 2021), and GSM8K (Cobbe et al., 2021), with the orchestrator evaluated on a held-out subset of the test set. For orchestration, initial

Table 2: **The orchestrator learns to start a chain more effectively as the training progresses.** This temporal shift in the distribution proves that the orchestrator is learning to identify specific experts that serve as stable initializers

| Expert | Epoch 1 | Epoch 2 | Epoch 3 | Epoch 4 | Epoch 5 |
|--------|---------|---------|---------|---------|---------|
| 1 | 0.064 | 0.045 | 0.006 | 0.036 | 0.065 |
| 2 | 0.169 | 0.227 | 0.163 | 0.078 | 0.039 |
| 3 | 0.158 | 0.085 | 0.157 | 0.071 | 0.354 |
| 4 | 0.204 | 0.201 | 0.086 | 0.096 | 0.020 |
| 5 | 0.029 | 0.036 | 0.048 | 0.085 | 0.041 |
| 6 | 0.126 | 0.177 | 0.081 | 0.126 | 0.053 |
| 7 | 0.003 | 0.010 | 0.053 | 0.031 | 0.055 |
| 8 | 0.068 | 0.007 | 0.192 | 0.186 | 0.303 |
| 9 | 0.046 | 0.029 | 0.063 | 0.117 | 0.024 |
| 10 | 0.128 | 0.181 | 0.144 | 0.166 | 0.044 |

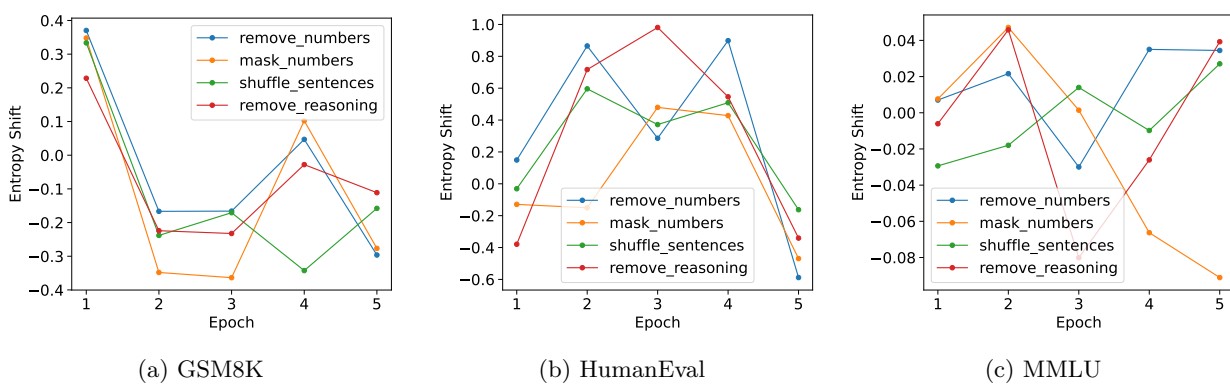

(a) GSM8K  (b) HumanEval  (c) MMLU

Figure 3: **Testing what the orchestrator actually cares about.** The plots show how routing entropy shifts when we damage the input. (a) On GSM8K, removing numbers (blue) causes the biggest reaction, confirming the model relies on numerical tokens. (b, c) On HumanEval and MMLU, shuffling sentences disrupts the model more, showing it depends on structural and semantic cues.

30 tokens generated by each experts is used as an input to the orchestrator. As the orchestrator learns to orchestrate, the performance on these tasks improve (see Table 8). Table 4 lists the hyperparameters used for all runs. Appendix D mentions the decoding configurations. Prompt templates and more plots are provided in Appendices E and G respectively. To verify the generalizability of our findings beyond equal-capacity models, we also construct a secondary *heterogeneous consortium* comprising 10 experts with widely varying parameter counts from three instruction-tuned model families – LLaMA-3.2 1B (Grattafiori et al., 2024), Qwen2.5 3B (Yang et al., 2025b), and Mistral 7B (Jiang et al., 2023a). Unlike the primary setup where experts share a similar capacity, this configuration tests orchestration dynamics across models ranging from 1B to 7B and utilizes the same temperature variations (see Table 6b) to assess behavior in a mixed-capability environment. Unless explicitly stated otherwise, all reported results and analyses focus on the primary *homogeneous* consortium. A discussion of applicability of the `INFORM` framework on black-box and API-based systems is provided in Appendix K.

## 4.1 Effect of Prompt Perturbations on Orchestrator Behavior

If an orchestrator truly conditions routing on task-relevant semantics, controlled prompt perturbations should produce structured, interpretable changes in expert selection and routing confidence. In contrast, heuristic orchestration would show either undue sensitivity to irrelevant changes or insensitivity to the removal of critical information. We therefore assess robustness and routing sensitivity via systematic prompt perturbations.

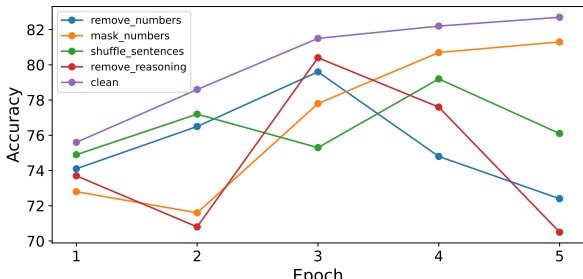 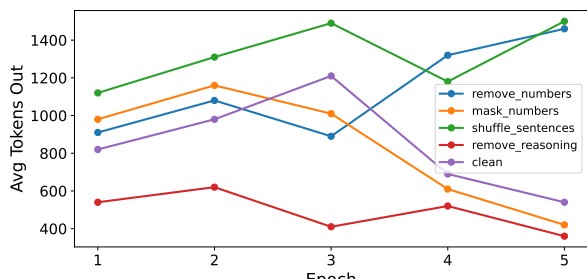

Figure 4: **Performance over Training Epochs on MMLU with Perturbations**. Clean data yields stable accuracy gains. In contrast, structural perturbations (e.g., removing numbers or reasoning) induce severe volatility and sharp late-epoch degradation, exposing the policy's brittleness to disrupted semantic cues.

Figure 5: **Token Costs over Training Epochs on MMLU with Perturbations**. The clean baseline trends toward concise outputs in later epochs as certainty solidifies. Conversely, disruptive perturbations like shuffling sentences trigger progressive verbosity, reflecting elevated policy entropy and uncertainty as the model struggles with input ambiguity.

**Perturbation Design.** For each task, we construct four classes of semantically targeted perturbations: (i) *removal of numerical tokens*, (ii) *masking of numerical tokens*, (iii) *sentence-level shuffling*, and (iv) *removal of explicit reasoning cues*. These perturbations are chosen to selectively disrupt task-relevant information without introducing distributional artifacts. For each perturbation, we measure changes in both the marginal sequence distribution and the conditional interaction matrix.

**Routing Sensitivity.** We quantify routing sensitivity as the KL divergence between the baseline and perturbed sequence distributions. Figure 3 shows that routing sensitivity is highest in early training epochs, indicating that the orchestrator initially relies on shallow prompt cues. As training progresses, sensitivity becomes more structured and perturbation-specific. In particular, perturbations on numbers induce larger shifts on GSM8K, while sentence shuffling and reasoning removal have a stronger effect on MMLU and HumanEval. This task-dependent sensitivity suggests that the orchestrator learns to associate specific prompt features with expert sequencing decisions rather than responding uniformly to surface-level changes.

**Routing Confidence under Perturbations.** To assess how perturbations affect decisiveness, we analyze changes in routing entropy. Early in training, perturbations frequently induce entropy collapse, reflecting brittle confidence shifts when salient cues are removed. Over training, the behavior becomes more structured but reveals a complex calibration profile. While semantic perturbations often increase entropy, we observe distinct failure modes where the orchestrator exhibits increased confidence under critical perturbations. These are analyzed in Appendix H. As shown in Figure 5, this elevated uncertainty manifests as progressive verbosity. While the clean baseline trends toward concise outputs in later epochs, structurally confusing perturbations, such as shuffling sentences drive token costs significantly higher, indicating the model is effectively 'rambling' as it struggles to resolve input ambiguity.

**Impact on Task Performance.** The consequences of these routing disruptions are directly reflected in task accuracy (Figure 4). While the clean baseline exhibits stable, monotonic performance gains across epochs, introducing structural perturbations exposes the policy's brittleness. Destructive alterations, such as removing numbers or explicit reasoning cues, induce high volatility and sharp late-epoch performance degradation, demonstrating that the model fails to maintain accurate generation when critical semantic cues are compromised.

**Emergent Robustness.** In later epochs, the orchestrator becomes more robust: routing is stable under mild perturbations yet responsive to semantically destructive ones, indicating grounding in meaningful features rather than brittle lexical cues. This robustness emerges gradually and unevenly across tasks, highlighting the need for training-aware analysis of orchestration behavior.

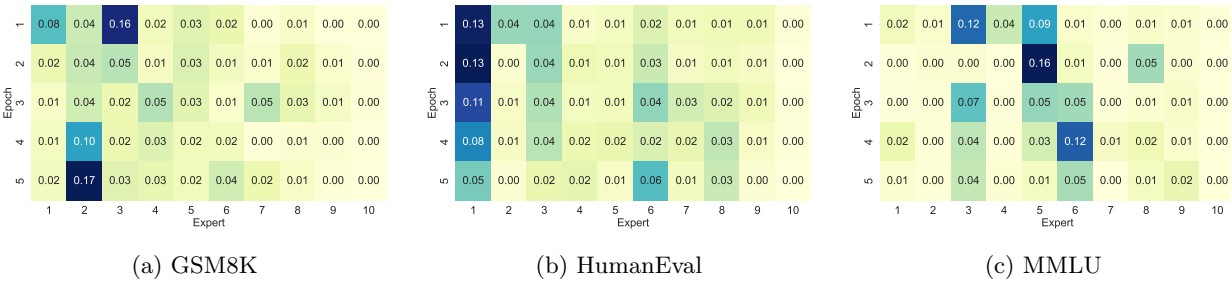

| | (a) GSM8K | (b) HumanEval | (c) MMLU |

Figure 6: **Intrinsic Expert Importance (Gradient Attribution) for Homogeneous Consortium.** These heatmaps visualize which experts exhibit the highest local sensitivity. Darker cells indicate experts with higher gradient norms, which means that the orchestrator relies heavily on their internal representations. The sparse patters reveal that only a small subset of experts are actually necessary.

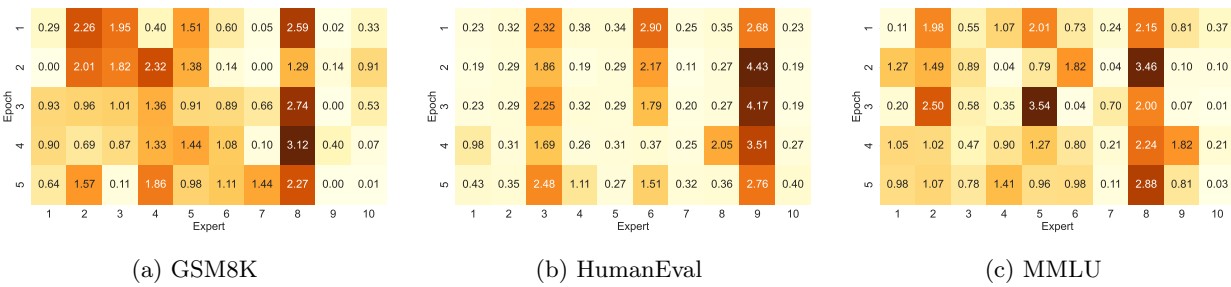

| | (a) GSM8K | (b) HumanEval | (c) MMLU |

Figure 7: **Relational Importance (Incoming Routing Mass) for Homogeneous Consortium.** These heatmaps show which experts get selected the most often by others. By comparing this to Figure 6, we can see alignment gaps, i.e., experts that appear popular here (high routing mass) but were not as important in Figure 6 (low gradient influence).

These results show that prompt perturbation analysis serves as a sensitivity probe of orchestrator behavior, revealing not just sensitivity but growing alignment between expert sequencing and task-relevant semantics. This underscores the need for interpretable orchestration frameworks that explain not only expert selection but why it remains stable or fragile under input variation.

## 4.2 Expert Attribution and Importance

Aggregate routing statistics reveal which experts are frequently selected, but they do not explain *why* the orchestrator prefers particular experts. To establish functional attribution, we analyze how expert-specific internal representations influence expert selection decisions, and how this intrinsic influence relates to realized routing behavior. We explicitly distinguish between *intrinsic expert importance*, measured via gradient-based attribution, and *relational importance*, measured via routing mass.

**Intrinsic Attribution via Gradient Sensitivity.** We estimate intrinsic expert importance (see Figure 6) by measuring the sensitivity of selection logits to perturbations in expert representations, computed as the norm of each expert's logit gradient with respect to its shared representation and aggregated across inputs. This yields a functional measure of an expert's influence on selection, capturing internal computational dependence rather than usage frequency. Intrinsic attribution is sparse across tasks (see Figure 17), with only a small subset of experts exerting consistent influence, and these experts vary by task, indicating task-dependent importance rather than fixed expert utility.

**Relational Importance via Routing Mass.** In parallel, we quantify relational importance (see Figure 7) using the total incoming routing mass derived from the conditional interaction matrix. This captures how frequently an expert is selected as a successor by other experts, reflecting its structural position within the

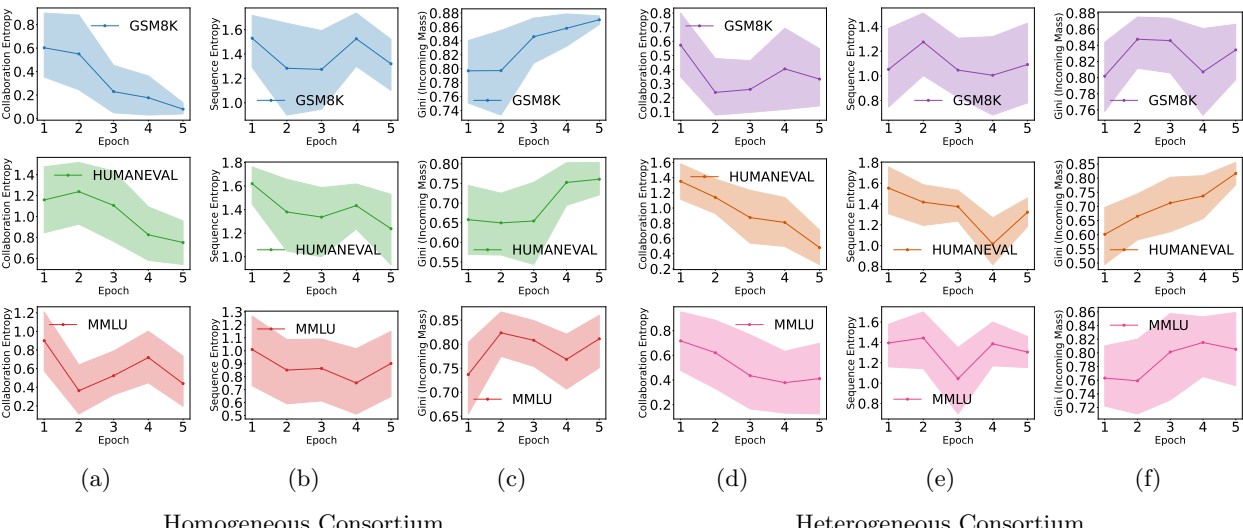

Figure 8: **Collaboration entropy, sequence entropy, and expert centralization over training for *homogeneous* (a-c) and *heterogeneous* (d-f) consortia.** Training shifts routing from exploratory to confident, with decreasing collaboration entropy and increased centralization. Ordering entropy remains nonzero, indicating soft, adaptive sequencing. Relative to the homogeneous setting, heterogeneous dynamics are less stable and more task-dependent, with slower convergence and weaker centralization. We discuss more about the behaviour of the heterogeneous consortium in Section 5.

collaboration graph. As training progresses, routing mass becomes increasingly skewed, with a small number of experts emerging as dominant successors. However, high routing mass alone does not imply intrinsic importance. Several experts receive substantial routing probability despite weak gradient attribution, suggesting that they are selected due to interaction-level dependencies rather than standalone representational influence. Additional visualizations are provided in Figure 18.

**Alignment and Misalignment.** Comparing intrinsic attribution and relational importance shows task-dependent alignment (Table 9). In GSM8K, high-attribution experts also receive high routing mass, indicating grounding in intrinsic competence. HumanEval shows partial alignment, with some dominance driven by interaction effects beyond intrinsic strength. MMLU exhibits the weakest alignment, reflecting domain heterogeneity and strong context-dependent complementarity. These results show that routing dominance does not imply intrinsic importance – only when intrinsic attribution and relational importance align is an expert structurally essential to orchestration.

## 4.3 Emergence of Routing Confidence and Centralization

Beyond expert attribution, effective orchestration requires resolving global coordination properties such as decisiveness and specialization. We analyze how routing confidence and successor centralization evolve during training in Figure 8.

**Routing Confidence.** We measure routing confidence as the entropy of the conditional interaction matrix, aggregated across inputs. Lower entropy corresponds to sharper, more decisive routing. As shown in Figure 8a routing entropy decreases consistently over training across tasks, indicating a transition from exploratory to confident expert selection. This decrease is most pronounced on GSM8K, where routing decisions stabilize early. HumanEval has a gradual decline, MMLU exhibits a non-monotonic behavior, which hints at sustained uncertainty due to domain variation.

**Successor Centralization.** To quantify specialization, we compute the Gini coefficient over incoming routing mass. A high value indicates that routing probability concentrates on a small subset of experts. We

observe consistent growth in centralization across tasks (Figure 8c). The asynchronous rise in centralization reflects the model adhering to the sparsity curriculum due to an adaptive-$k$ schedule (Appendix C) before fully resolving the optimal routing pathways.

**Decoupling of Confidence and Centralization.** Importantly, comparing Figures 8a, and Figure 8c reveals that routing confidence and centralization do not evolve synchronously. In several cases, centralization increases before routing entropy fully stabilizes, indicating that the orchestrator may first learn *who* to trust before resolving *how confidently* to route. This decoupling underscores the need to analyze orchestration along multiple dimensions rather than through a single scalar metric.

## 4.4 Emergence of Expert Ordering

In sequential orchestration, the order in which experts are invoked plays a critical functional role, as early-stage experts condition the intermediate representations seen by all downstream experts. To study ordering behavior, we analyze entropy of marginal initial selection distribution.

**Ordering Entropy.** We measure ordering entropy as the entropy of the marginal distribution over the first selected expert. High entropy corresponds to diffuse ordering preferences, while low entropy indicates strong preference for specific initial experts. Across all tasks, ordering entropy decreases over training but remains significantly above zero (Figure 8b), indicating that the orchestrator learns structured yet non-deterministic ordering preferences. GSM8K exhibits the strongest reduction in ordering entropy, suggesting the emergence of a preferred reasoning initializer. In contrast, HumanEval and MMLU retain higher entropy, reflecting the existence of multiple viable entry points into the reasoning chain. This indicates that sequencing is learned as a soft constraint rather than a rigid policy.

**Interaction with Attribution and Centralization.** Notably, ordering preferences do not always align with intrinsic expert attribution or successor centralization. Some experts are frequently selected as initializers despite modest intrinsic attribution, suggesting that suitability for early-stage reasoning is not solely determined by standalone expert strength. This further motivates separating ordering analysis from attribution and routing dynamics.

## 4.5 Ablation Study and Analysis

We perform the following ablations to assess the structural orchestration components: (i) replacing the learned, input-conditioned collaboration matrix with a static uniform graph (Figure 14a), (ii) fixing the inference to a static execution sequence (Figure 14b), (iii) masking the single most intrinsically important expert identified by gradient-based attribution (Table 7, Figure 11), (iv) We further analyze the contribution of probabilistic routing components in Figure 12, which contrasts the full INFORM model against *Relational-Only* and *Intrinsic-Only* variants. We discuss about failure modes of orchestration in Appendix H. Appendix F shows that adaptive interaction structure and dynamic sequencing are both necessary for strong performance, and that experts with high intrinsic attribution tend to exert disproportionate influence on routing (Tables 7, 10).

# 5 Discussion

**How does a *heterogeneous consortium* of experts affect orchestration stability?** To validate whether our findings hold across diverse model sizes and capabilities, we analyze the orchestration dynamics of the heterogeneous consortium and contrast them with the homogeneous consortium (Figure 8). While the homogeneous setting is characterized by a rapid, monotonic transition to confident routing and strong successor centralization, the heterogeneous environment, spanning 1B to 7B parameter models, introduces significant volatility. As observed in Figure 8d and 8f, routing confidence converges more slowly, and centralization is notably weaker, particularly on broad-domain tasks like MMLU where the Gini coefficient remains fluctuating. We hypothesize that this *instability* reflects the orchestrator's struggle to resolve the complex trade-off between expert capability and specialization. Unlike the homogeneous case where experts

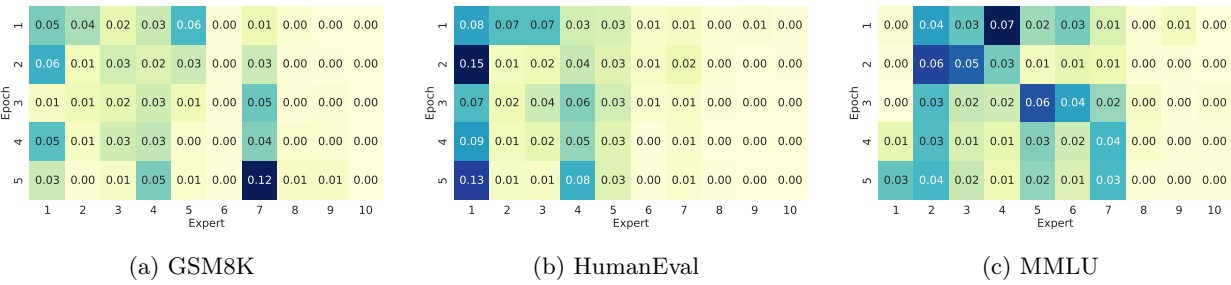

|          | (a) GSM8K | (b) HumanEval | (c) MMLU |
|----------|-----------|---------------|----------|

Figure 9: **Intrinsic Expert Importance (Gradient Attribution) for Heterogeneous Consortium.** In contrast to the highly concentrated reliance observed in the homogeneous setup, these heatmaps reveal a significantly more distributed pattern of expert utilization. While the orchestrator continues to favor specific models depending on the task and epoch, the gradient norms indicate that it draws upon a wider, more diverse subset of experts simultaneously, resulting in a notably less sparse distribution of importance.

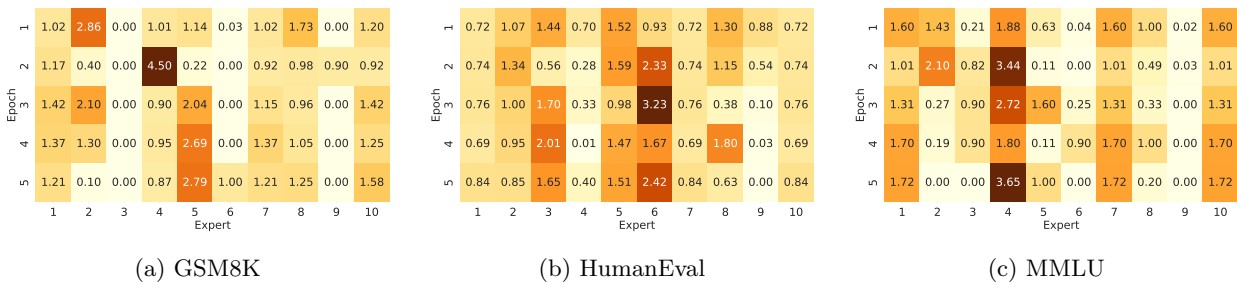

|          | (a) GSM8K | (b) HumanEval | (c) MMLU |
|----------|-----------|---------------|----------|

Figure 10: **Relational Importance (Incoming Routing Mass) for Heterogeneous Consortium.** Similar to the homogeneous setup, these heatmaps reveal persistent alignment gaps where frequently selected experts do not always possess high gradient influence. However, the heterogeneous routing behavior is noticeably more polarized, with incoming routing mass clustering sharply on a select few experts rather than being broadly scattered across the consortium.

are capacity-equivalent, the heterogeneous setting forces the orchestrator to dynamically arbitrate between the raw reasoning power of larger models (e.g., Mistral 7B) and the potential sufficiency of smaller experts (e.g., LLaMA 3.2 1B). The policy retains higher entropy (Figure 8e), indicating a softer, more adaptive sequencing strategy that resists collapsing into the static patterns seen in the homogeneous setup. This distributed reliance is further supported by the intrinsic expert importance (Figure 9), which demonstrates that gradient attribution is spread across a significantly wider, more diverse subset of models compared to the sparse utilization in the homogeneous baseline. Interestingly, despite this broad intrinsic reliance, the orchestrator's routing mechanism exhibits polarized behavior, with incoming routing mass sharply clustered around a select few models (Figure 10). This contrast highlights a persistent alignment gap in the heterogeneous environment, demonstrating that the experts most frequently selected by the routing policy do not necessarily possess the highest underlying gradient influence.

**Why is learned orchestration better than baselines with only structured interactions?** To validate the efficiency of INFORM's dynamic routing against rigid multi-agent frameworks, we compare it with MetaGPT (Hong et al., 2024) using a fixed consortium of role-specific experts. We select MetaGPT as the representative baseline because it establishes the state-of-the-art for rigid, purpose-driven coordination, offering the clearest contrast to learned orchestration. As detailed in Table 3, we assign models of identical sizes from the *homogenous consortium* to different roles to keep underlying expert capabilities constant. INFORM consistently reduces interaction overhead relative to MetaGPT's Standard Operating Procedures, with the largest gains in complex roles: *Engineer* calls drop to 0.69 and *Architect* calls drop to 0.32. We hypothesize that while MetaGPT enforces predefined and often redundant communication loops, INFORM's learned policy identifies the minimal functional trajectory required to solve the task, pruning unnecessary

Table 3: **Average number of model calls and performance on HumanEval.** MetaGPT invokes all roles once per instance by design, while our orchestrator adaptively selects experts, resulting in substantially fewer model calls. This comparison isolates adaptive orchestration efficiency rather than architectural parity.

| Method | Engineer | QA | Architect | PM | ProjM | Avg. Calls | Pass@1 |
|--------|----------|------|-----------|------|-------|------------|--------|
| MetaGPT | 1.00 | 1.00 | 1.00 | 1.00 | 1.00 | 5.00 | 85.9 |
| INFORM | 0.69 | 0.26 | 0.32 | 0.10 | 0.07 | **1.44** | **87.1** |

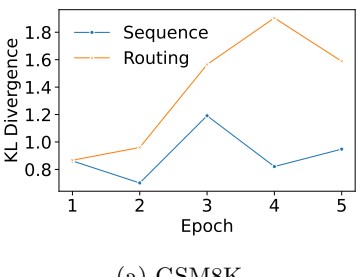

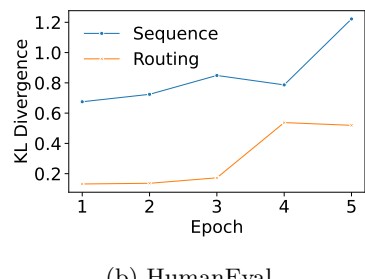

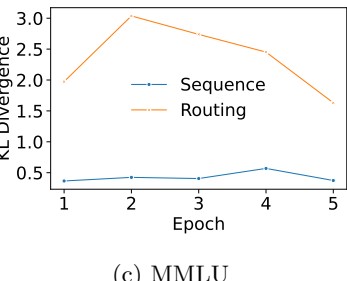

(a) GSM8K
(b) HumanEval
(c) MMLU

Figure 11: **Effect of masking the most intrinsically important expert.** KL divergence of sequence and interaction distributions across training. (a, c) On GSM8K and MMLU, higher routing divergence indicates that key experts act as interaction hubs. (b) HumanEval instead shows higher sequencing divergence, reflecting sensitivity to initialization.

expert invocations and yielding a leaner, more cost-effective pipeline. On HumanEval, this translates to a $\sim 3.5\times$ speedup with a $\sim 1.4\%$ performance gain. Importantly, this advantage is not limited to learned routing graphs: applying `INFORM` to confidence-based cascade orchestration (Chen et al., 2024) reveals a similar misalignment between stopping dominance and functional necessity (Appendix I), indicating that `INFORM`'s interpretability principles generalize across fundamentally different orchestration regimes.

**How does masking out the best-selected expert affect the attributions?** To probe the role of intrinsically important experts, we mask the single highest-ranked expert (by gradient-based intrinsic attribution) and measure the resulting changes in both the sequence and routing distributions. Figure 11 reports the induced KL divergence across training epochs for GSM8K, HumanEval, and MMLU. On GSM8K and MMLU (Figure 11a,c), masking induces substantially larger divergence in the routing distribution than in sequencing, indicating that dominant experts primarily act as *interaction hubs*: their removal minimally affects initial selection but strongly disrupts successor transitions and expert dependencies. This effect is most pronounced on MMLU, where routing divergence remains consistently high, suggesting that broad-domain reasoning relies on a stable interaction topology anchored by a small number of central experts. In contrast, HumanEval exhibits the opposite behavior (Figure 11b) as masking the top expert produces higher divergence in the sequence distribution, indicating that expert importance is concentrated at the *initial selection* stage, while downstream interactions remain comparatively robust. These results show that expert importance is task-dependent – reasoning-heavy benchmarks depend on interaction stabilization, whereas code generation relies on precise expert initialization.

**What drives effective expert routing: structure or semantics?** To disentangle the contributions of interaction history and instance-specific scoring, we analyze the probabilistic components governing expert selection in Figures 12 and 13. We compare the full orchestration setup with three variants: a *Uniform Baseline*, a *Relational-Only* setting driven by the transition matrix $\mathbf{C}(x)$, and an *Intrinsic-Only* setting governed solely by the sequence distribution $s(x)$. Across all tasks, the uniform baseline performs substantially worse, confirming that gains arise from learned routing rather than ensemble size. *Relational-Only Orchestration* exhibits strong successor centralization on GSM8K and MMLU (Gini $\geq 0.80$), indicating over-reliance on interaction hubs; while effective on GSM8K, it underperforms on MMLU and HumanEval, where semantic sensitivity is critical. In contrast, *Intrinsic-Only Orchestration* reduces centralization and yields large gains

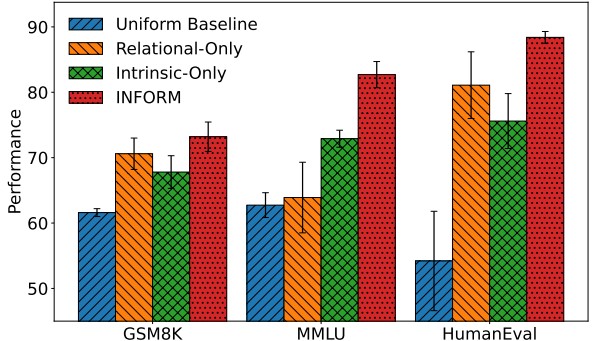

Figure 12: **Task performance under routing ablations.** Accuracy on GSM8K, MMLU, and Pass@1 on HumanEval, for the full model and ablations. Removing relational structure or instance-specific intrinsic scoring degrades performance, with the largest drops on heterogeneous tasks like MMLU.

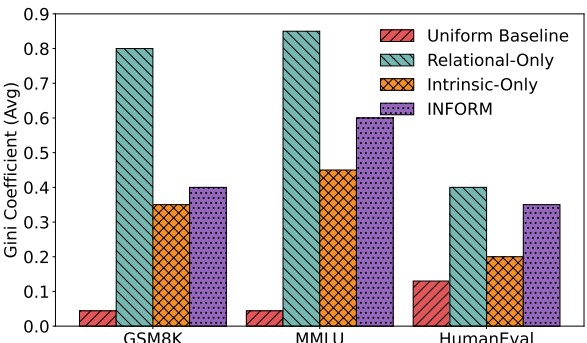

Figure 13: **Routing centralization from orchestration signals.** Average Gini coefficients of expert selection across tasks. Relational-only routing is highly centralized, intrinsic-only routing is more uniform, and the full model balances structural consistency with semantic adaptivity.

on MMLU (+9.0%) and HumanEval, demonstrating the importance of content-aware expert selection. The full `INFORM` setup achieves the best performance across all tasks by integrating both signals, maintaining moderate centralization (Gini ≈ 0.35–0.60) and balancing structural stability with semantic precision.

## 6 Conclusion

In this work, we introduce `INFORM`, an interpretability framework that treats neural orchestration as an explicit, analyzable computation, enabling the decoupling of expert interaction, sequencing, and intrinsic attribution. Our results show that orchestration converges to structured collaboration, but that routing frequency often misrepresents functional necessity – experts that dominate routing are not always those that are functionally required. By revealing systematic divergence between intrinsic gradient-based attribution and relational routing mass, `INFORM` exposes hidden structural dependencies and interaction hubs that are invisible to accuracy-based evaluation alone. While our gradient-based approach captures local sensitivity of routing decisions rather than full causal structures, these targeted interventional signals remain highly effective for diagnosing system behavior. We demonstrate that orchestration dynamics emerge asynchronously, with centralization and trust in specific experts often preceding stable routing confidence, and with sequencing remaining deliberately non-deterministic across tasks. Together, these findings highlight the importance of interpretability-driven analysis for diagnosing brittleness, redundancy, and failure propagation in multi-expert systems, and position `INFORM` as a practical tool for understanding and improving learned orchestration beyond what performance metrics alone can capture.

**Scope of Analysis.** The scope of our analysis encompasses the analysis of multi-expert orchestration by decoupling interaction structure, sequencing, and intrinsic attribution. We validated the framework on a differentiable orchestration architecture with a lightweight MLP-based orchestrator, demonstrating its versatility by comparing its learned routing against the rigid workflows of MetaGPT and successfully adapting its intrinsic importance metrics to evaluate confidence-based FrugalGPT-style cascades. While extracting full attribution requires access to the orchestrator, the constituent experts themselves can remain entirely black-box, allowing practitioners to leverage `INFORM`'s structural analyses and perturbation sensitivity to reliably diagnose system dependencies and interaction hubs.

**Limitations.** A primary limitation is that our intrinsic importance metric relies on gradient-based attribution, which captures local computational sensitivity rather than establishing a formal causal graph or interventional guarantees. Additionally, computing these exact causal signals strictly requires white-box access to the orchestrator's internal representations, logits, and gradients, making the complete framework

inapplicable to fully opaque, API-driven routing policies. Finally, the framework's generalizability remains untested in fundamentally different paradigms, such as fully decentralized multi-agent topologies, emergent communication networks, or orchestration driven entirely by reinforcement learning.

**Acknowledgments**

The authors acknowledge the financial support of DYSL-AI, India. Tanmoy Chakraborty acknowledges the support of the Google GCP Grant and the Rajiv Khemani Young Faculty Chair Professorship in Artificial Intelligence.

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

## Structure of the Appendix

The appendix comprises of the following parts:

**Appendix A: Frequently Asked Questions and Key Takeaways.** We clarify design choices and scope, address common points of confusion, and summarize key insights from the presented analyses.

**Appendix B: Extended Related Work.** We contextualize our work within dynamic routing, multi-agent coordination, and interpretability research.

**Appendix C: Training the Orchestrator.** We provide details of the optimization objective, loss components, and hyperparameter configurations.

**Appendix D: Decoding Configurations.** We list the specific models used and temperature settings used for the expert consortium.

**Appendix E: Prompts.** We provide the prompt template used for expert coordination.

**Appendix F: Ablation Studies.** We evaluate the necessity of adaptive collaboration, dynamic sequencing, and intrinsic expert importance.

**Appendix G: Additional Results.** We present training dynamics, visualization of expert attributions over time, overhead analysis and detailed masking results.

**Appendix H: Observed Failure Modes.** We categorize systematic orchestration failures such as hub over-centralization and routing-attribution misalignment.

**Appendix I: Interpreting Cascade-based Orchestration.** We analyze confidence-based fixed-order cascade-style orchestration using intrinsic importance and functional dependency analyses.

**Appendix J: Statistical Tests and Quantitative Analyses.** We report statistical tests supporting our claims, including rank correlation and paired comparisons.

**Appendix K: Applicability to Black-Box and API-Based Systems.** We discuss about the applicability of our analysis and the INFORM framework for black-box and API-based systems.

**Appendix L: Individual Expert Baseline Scores.** We present the individual scores of each expert in the ten-expert homogeneous consortium.

**Appendix M: Robustness of the Intrinsic-Importance Metric.** We experiment and present the results of using alternative encoders and pooling strategies and whether it affects the ranking of experts.

**Appendix N: Prompts Used for Comparison with MetaGPT.** We provide the exact prompts and setup used to compare our canonical setup with MetaGPT.

## A   Frequently Asked Questions and Key Takeaways

This section addresses common questions and potential points of confusion regarding the INFORM framework, its scope, and the interpretation of our results, and summarizes key takeaways from our analyses. The goal is to clarify design choices, distill central insights, and prevent misinterpretation of attribution and orchestration analyses.

## A.1 Frequently Asked Questions

**Q1: Is INFORM a causal method in the formal sense?** INFORM does not claim to recover causal structure in the sense of formal causal graphs or interventional guarantees as defined in classical causality frameworks. Instead, it provides *attribution-based signals* derived from gradient-based sensitivity and targeted interventions. These signals identify functional dependence of routing decisions on expert representations, which is sufficient for diagnosing structural dependencies and failure propagation in orchestration, but should not be interpreted as establishing full causal mechanisms.

**Q2: Why do we use gradient-based attribution for expert importance?** Gradient-based attribution is a natural choice because the orchestrator is a differentiable controller. Gradients measure local sensitivity of selection logits to expert representations, directly capturing whether changes in an expert's internal state would affect routing decisions. Unlike routing frequency, which reflects realized usage, gradient attribution measures internal computational reliance. We emphasize that attribution scores reflect *influence on routing*, not expert correctness or output quality.

**Q3: Does high intrinsic importance mean an expert is better or more accurate?** No. Intrinsic importance measures how strongly an expert's representation influences the orchestrator's decisions, not the semantic quality or correctness of the expert's outputs. An expert can be intrinsically important because it anchors interaction structure or conditions downstream routing, even if it is not the most accurate standalone model.

**Q4: Why do some frequently routed experts have low intrinsic attribution?** This reflects the distinction between *relational* and *intrinsic* importance. Some experts function as interaction hubs due to favorable positioning in the collaboration graph or historical co-occurrence patterns, rather than because their internal representations are functionally important. INFORM explicitly separates these effects to avoid conflating popularity with necessity.

**Q5: How should routing entropy be interpreted?** Routing entropy measures the uncertainty of expert-to-expert transitions. High entropy indicates exploratory or uncertain routing, while low entropy indicates confident, concentrated selection. Importantly, low entropy is not always desirable: overconfident routing under semantically damaged inputs is a failure mode identified by our analysis. Routing entropy should therefore be interpreted jointly with perturbation sensitivity and attribution alignment.

**Q6: Why does centralization sometimes increase before routing confidence stabilizes?** This reflects an asynchronous learning dynamic. The orchestrator often first learns *which* experts tend to be useful (centralization) before learning *how confidently* to route between them (entropy reduction). This decoupling explains why expert dominance can emerge even while routing remains uncertain, particularly in heterogeneous tasks such as MMLU.

**Q7: Can INFORM be applied to black-box or API-based systems?** The current formulation of INFORM requires white-box access to the orchestrator to compute gradient-based attribution and inspect interaction matrices. While some concepts, like routing frequency or perturbation sensitivity, may be approximated in black-box settings, full intrinsic attribution is not directly applicable without access to internal representations and gradients.

**Q8: Does masking an intrinsically important expert always reduce performance?** Not necessarily. In later training stages, redundancy among experts can partially compensate for the removal of an important expert, leading to smaller-than-expected drops in accuracy. However, even in such cases, routing analysis reveals significant structural disruption, such as increased routing KL divergence or rerouting through weaker interaction pathways. This highlights why accuracy alone is insufficient to assess robustness.

**Q9: Why does HumanEval behave differently from GSM8K and MMLU?** HumanEval is particularly sensitive to early-stage expert selection due to the syntactic and structural constraints of code

generation. As a result, sequencing effects dominate routing effects, and failures often arise from poor initialization rather than downstream interaction collapse. In contrast, GSM8K and MMLU rely more heavily on sustained expert interaction, making routing structure and interaction hubs more critical.

**Q10: What does `INFORM` enable that accuracy metrics cannot?**  Accuracy metrics reveal whether a system succeeds, but not *how* or *why*. `INFORM` exposes hidden dependencies, brittle coordination, redundancy, and failure propagation in multi-expert systems. It enables diagnosis of structural weaknesses before they manifest as accuracy degradation, and provides tools for principled expert pruning, regularization, and robustness analysis.

**Q11: How should practitioners use `INFORM` in practice?**  `INFORM` is best used as a diagnostic layer alongside performance evaluation. Practitioners can monitor attribution–routing alignment, detect over-centralization, identify critical interaction hubs, and evaluate robustness through targeted perturbations. These insights can guide architectural decisions, training regularization, and deployment safeguards without modifying the underlying experts.

### A.2  Key Takeaways

**Intrinsic Importance versus Routing.**  Routing dominance often diverges from intrinsic importance, as frequently selected experts are not always the most influential under the attribution metric. Gradient-based attribution reveals sparse intrinsic importance, whereas routing mass captures relational interaction hubs and structural positioning within the collaboration graph.

**Emergent Structure and Specialization.**  Orchestration shifts from exploratory routing to structured collaboration in training, evidenced by clear vertical banding in the collaboration matrix. This reflects the emergence of universal successors and a shrinking effective expert set. As routing confidence increases, centralization rises, though task-specific dynamics differ: GSM8K stabilizes early with strong successor dominance, HumanEval centralizes more gradually, and MMLU exhibits non-monotonic behavior.

**Sequencing and Ordering**  The orchestrator learns sequencing by identifying preferred initializers: starting-expert distributions sharpen from dispersed to peaked over training. Ordering remains non-deterministic, entropy decreases but stays above zero, indicating soft, adaptive preferences rather than rigid policies.

**Functional Grounding.**  Targeted masking of the highly attributed expert triggers routing collapse, supporting the alignment between intrinsic attribution and observed perturbation effects. Masking disrupts routing (expert–expert interactions) rather than sequencing patterns, highlighting the importance of interaction structure in orchestration.

## B  Extended Related Work

**Dynamic Routing and Mixture-of-Experts**  Conditional computation has long been studied as a mechanism for improving efficiency and specialization in neural networks. The Mixture-of-Experts (MoE) paradigm routes inputs to subsets of parameters via learned gating functions, enabling scalable training and inference (Shazeer et al., 2017; Fedus et al., 2022). While highly effective, MoE routing is typically optimized implicitly through task loss, with limited emphasis on interpreting routing decisions or understanding whether selected experts are necessary. More recent work extends routing beyond a single model, selecting among multiple frozen language models (Chen et al., 2024; Ong et al., 2025). Systems such as LLM-Blender (Jiang et al., 2023b) focus on ranking or aggregating candidate outputs, implicitly assuming that expert contributions are exchangeable. While both MoE and external routing frameworks learn complex selection policies, they provide limited visibility into why specific experts are preferred, how dependencies arise, or whether routing frequency reflects expert importance. In contrast, our analysis highlights that orchestration is inherently sequential and routing decisions encode temporal structure that cannot be captured by static aggregation alone.

**Multi-Agent Collaboration and Coordination**   Recent surveys categorize multi-agent systems based on how they interact, ranging from centralized pipelines to decentralized networks with emergent behaviors (Tran et al., 2025; Guo et al., 2024). Frameworks such as MetaGPT enforce rigid workflows, while others allow more flexible interaction patterns (Hong et al., 2024). However, decentralized collaboration introduces vulnerability to faulty or misleading agents, where errors can propagate across the system (Huang et al., 2025). Instead, INFORM focuses on diagnosis of failure propagation and expert influence without modifying the collaboration protocol itself.

**Robustness, Failure Propagation, and Expert Reliability.**   A growing field of work studies robustness in multi-agent and ensemble-based systems, examining how adversarial or faulty agents impact system-level performance (Huang et al., 2025; Vallecillos-Ruiz et al., 2025). These studies demonstrate that even a single unreliable expert can induce disproportionate degradation, particularly when errors propagate silently. While prior work quantifies robustness at the system level, orchestration itself is typically treated as fixed or implicit. As a result, it remains unclear which experts are functionally critical, how redundancy is exploited, or how routing adapts under perturbation. Through our perturbation and ablation analyses in later sections, we examine how existing orchestration policies respond to disruptions, revealing when decisions are semantically grounded versus brittle.

**Interpretability of Controllers**   Interpretability research has made substantial progress in understanding Transformer-based models through attention visualization, probing classifiers, and attribution methods (Vig, 2019; Clark et al., 2019; Conneau et al., 2018). More recent work advocates for mechanistic and circuit-level analysis to move beyond correlational explanations (Olah et al., 2020).

Existing work either optimizes orchestration without interpretability, studies collaboration without auditing influence, or analyzes model internals without addressing expert-level control. INFORM unifies these perspectives by enabling direct analysis of attribution, interaction, and sequencing in learned orchestration policies.

## C   Training the Orchestrator

We use a composite optimization objective $\mathcal{L}$ to train the orchestrator in presence of an oracle LLM. We use GPT OSS 20B (Agarwal et al., 2025) as the oracle LLM across all training runs. The orchestrator uses a frozen BERT-based encoder (Devlin et al., 2019) that produces 768-dimensional embeddings. The components of the composite objective function are defined as follows:

- **Utility** $\mathcal{L}_{\text{utility}}$ (task loss on $y_K$ vs. $y^*$): anchors training to the end-task so that all collaboration ultimately improves the *final* prediction, not merely intermediate consistency.

- **Distillation** $\mathcal{L}_{\text{distill}}$ $\|o - o_{\mathcal{E}_K}\|_2^2$: encourages the final expert to mimic the oracle's reasoning by minimizing the mean-squared error between their respective hidden representations. This component acts as a stabilizer and helps accelerate convergence, especially for small consortia.

- **Symmetry** $\mathcal{L}_{\text{symm}} = \|C - C^\top\|_F^2$: discourages brittle, one-way topologies in the collaboration graph. Near-symmetric $C$ promotes *reciprocal* information flow, which empirically yields shorter, more stable refinement chains and better robustness to population changes (e.g., when an expert is removed or replaced). Ablations confirm this is a critical structural prior, preventing performance degradation in larger consortia.

- **Sparsity** ($\mathcal{L}_{\text{spar}}$): encourages a sparse collaboration matrix by penalizing the average entropy of its rows. For each row $C_i$ (distribution of expert $i$'s outgoing weights), we compute the entropy $H(C_i) = -\sum_j C_{i,j} \log C_{i,j}$, and penalize its average across all experts. This encourages each expert to rely on a small, selective set of other experts rather than forming diffuse connections.

- **Oracle alignment** $\mathcal{L}_{\text{oracle}} = \frac{1}{M^2} \sum_{i,j} (C_{ij} - C_{ij}^{\text{oracle}})^2 + \frac{1}{M} \sum_i (\pi_i - \pi_i^{\text{oracle}})^2$ with $C_{ij}^{\text{oracle}} = \cos\left(\frac{o_i + o_j}{2}, o\right)$ ($i \neq j$) and $\pi_i^{\text{oracle}} = \cos(o_i, o)$: bootstraps a sensible *initial protocol*. It steers both *who* to select and *how*

to connect toward oracle-indicated preferences, then cedes to the utility loss as training progresses. This prevents long cold-start phases where the orchestrator explores unproductive chains. Ablations confirm the oracle acts as an accelerator and stabilizer for early training, with test-time performance driven by the learned protocol.

- **Diversity** $\mathcal{L}_{\text{diver}} = -\frac{1}{M}\text{var}(s_1, \ldots, s_M)$ with $s_i = \sum_{k \in \text{TopK}(\pi)} \pi_k \mathbf{1}[k = i]$: counters *mode collapse* onto a single persistent expert by encouraging balanced utilization across the consortium. This broadens exploration, improves coverage on heterogeneous workloads, and works in tandem with sparsity to allocate distinct niches.

- **Selection entropy** $\mathcal{L}_{\text{sel}} = -\frac{1}{M}\sum_i \pi_i$: encourages *decisive* selections (low entropy), which reduces dithering across many near-tied experts, stabilizes the Gumbel-Softmax path, and shortens realized chains by clarifying the top-$K$.

- **Length penalty** $\mathcal{L}_{\text{len}} = K \cdot \alpha$: encodes the compute budget directly into the objective. It regularizes toward *short* chains, synergizing with early stopping and the length-aware score $f_{\text{len},e}$ to yield favorable accuracy-latency trade-offs.

Therefore, the complete objective to optimize is given by:

$$\begin{aligned}
\mathcal{L}_{\text{total}} &= \lambda_{\text{utility}}\mathcal{L}_{\text{utility}} + \lambda_{\text{distill}}\mathcal{L}_{\text{distill}} + \lambda_{\text{symm}}\mathcal{L}_{\text{symm}} + \lambda_{\text{spar}}\mathcal{L}_{\text{spar}} \\
&\quad + \lambda_{\text{oracle}}\mathcal{L}_{\text{oracle}} + \lambda_{\text{diver}}\mathcal{L}_{\text{diver}} + \lambda_{\text{sel}}\mathcal{L}_{\text{sel}} + \lambda_{\text{len}}\mathcal{L}_{\text{len}}
\end{aligned}$$

We train the orchestrator on a single NVIDIA A100 80GB GPU on the training sets of the publicly-available datasets. Table 4 summarizes the hyperparameters used across all experiments. A held-out set of 100 examples were used to determine the weights of the loss terms. The values for the coefficients of all loss terms have been determined after a coarse sweep through magnitudes $\{10^{-2}, 10^{-1}, 10^0\}$ and are reported in Table 5. The trainable routing and sequencing heads are optimized using the Adam optimizer and a cosine LR schedule with warmup.

**Adaptive Sparsity Schedule.** To induce the specialization observed in our results, we employ an Adaptive-Top-$k$ mechanism. Rather than relying solely on entropy penalties, we explicitly decay the number of active experts $k$ from $N$ to 1 based on a moving average of selection confidence, structurally forcing the orchestrator to transition from broad exploration to focused execution.

## D  Decoding Configurations

Table 6 reports the models and inference-time temperature configurations used in our experiments. By *homogeneous*, we imply models of similar capacities and model sizes. Conversely, models of different sizes and capabilities make up the *heterogeneous* consortia. All models are instruction-tuned. For DeepSeek-R1, we use the distilled `DeepSeek-R1-0528-Qwen3-8B` variant. The sampling parameters are set as follows across all tasks and expert calls – (i) Qwen3, Qwen2.5, DeepSeek-R1: temperature : 0.3, top_p : 0.7, (ii) LLaMA, Mistral, GPT-OSS-20B: temperature : 0.1, top_p : 0.7.

## E  Prompts

During inference, all experts are prompted using the following prompt template, regardless of the task being undertaken. The response of the last consulted expert is prepended to the prompt for each turn of the inference. A natural language formulation of the task is added at the end of the prompt.

```
Expert 1's Response: ...

Above is the conversation history, with the most recent model output at the top. Each
model should carefully read *all previous outputs* and decide how to contribute next.
Your role is to coordinate with earlier outputs by either:
```

Table 4: Hyperparameters for orchestrator training.

| Hyperparameter | Value |
|---|---|
| *Optimization* | |
| Learning Rate | $1 \times 10^{-3}$ |
| Batch Size | 2 |
| Training Epochs | 5 |
| Warmup Ratio | 0.1 |
| Gradient Clipping Norm | 15.0 |
| *Model Architecture* | |
| Hidden Dimension ($d$) | 256 |
| Attention Heads (Backbone) | 4 |
| Dropout | 0.1 |
| *Routing & Gumbel-Softmax* | |
| Initial Temperature ($\tau$) | 1.0 |
| Minimum Temperature ($\tau_{\min}$) | 0.5 |
| Temperature Decay ($\gamma$) | 0.999 |
| *Loss Coefficients* | |
| Selection Loss Weight ($\lambda_{\text{select}}$) | 1.0 |
| Oracle Alignment Weight ($\lambda_{\text{oracle}}$) | 0.5 |
| Utility Loss Weight ($\lambda_{\text{util}}$) | 0.5 |
| Distillation Loss Weight ($\lambda_{\text{distill}}$) | 0.5 |
| Length Penalty Weight ($\lambda_{\text{len}}$) | 0.5 |
| Symmetry Regularization ($\lambda_{\text{symm}}$) | 0.05 |
| Sparsity Regularization ($\lambda_{\text{sparse}}$) | 0.1 |
| Diversity Regularization ($\lambda_{\text{div}}$) | 0.1 |

Table 5: Hyperparameter sweep for selected $\lambda$ coefficients in the homogeneous setting with 5× Qwen3-8B experts.

(a) GSM8K

| Coefficient | $\lambda = 1.0$ | $\lambda = 0.1$ | $\lambda = 0.01$ |
|---|---|---|---|
| $\lambda_{\text{oracle}}$ | 87.8 | **91.0** | 87.1 |
| $\lambda_{\text{symmetry}}$ | 89.4 | **91.0** | 90.1 |
| $\lambda_{\text{sparsity}}$ | 87.9 | 87.9 | **91.0** |

(b) MMLU

| Coefficient | $\lambda = 1.0$ | $\lambda = 0.1$ | $\lambda = 0.01$ |
|---|---|---|---|
| $\lambda_{\text{oracle}}$ | 90.20 | **96.08** | 89.54 |
| $\lambda_{\text{symmetry}}$ | 89.41 | **96.08** | 90.20 |
| $\lambda_{\text{sparsity}}$ | 88.89 | 87.58 | **96.08** |

```
1. Building upon correct reasoning.
2. Correcting or refining mistakes.
3. Adding missing details.
4. Passing an intermediate or final answer if complete.
Always state explicitly what you are doing and why. Avoid repeating identical reasoning
unless you are clarifying or improving it. /no_think

{task_instance}
```

Table 6: Models, parameter scale, and decoding temperatures used in experiments.

(a) Homogeneous Consortium

| Expert | Model Family | Size | Temperature $\tau$ |
|---|---|---|---|
| 1 | LLaMA 3.1 | 8B | 8E-6 |
| 2 | Qwen3 | 8B | 0.2 |
| 3 | DeepSeek-R1 | 8B | 0.5 |
| 4 | LLaMA 3.1 | 8B | 0.5 |
| 5 | Qwen3 | 8B | 8E-6 |
| 6 | DeepSeek-R1 | 8B | 1.0 |
| 7 | LLaMA 3.1 | 8B | 8E-6 |
| 8 | Qwen3 | 8B | 1.5 |
| 9 | DeepSeek-R1 | 8B | 2.0 |
| 10 | LLaMA 3.1 | 8B | 8E-6 |

(b) Heterogeneous Consortium

| Expert | Model Family | Size | Temperature $\tau$ |
|---|---|---|---|
| 1 | LLaMA 3.2 | 1B | 8E-6 |
| 2 | Qwen2.5 | 3B | 0.2 |
| 3 | Mistral | 7B | 0.5 |
| 4 | LLaMA 3.2 | 1B | 0.5 |
| 5 | Qwen2.5 | 3B | 8E-6 |
| 6 | Mistral | 7B | 1.0 |
| 7 | LLaMA 3.2 | 1B | 8E-6 |
| 8 | Qwen2.5 | 3B | 1.5 |
| 9 | Mistral | 7B | 2.0 |
| 10 | LLaMA 3.2 | 1B | 8E-6 |

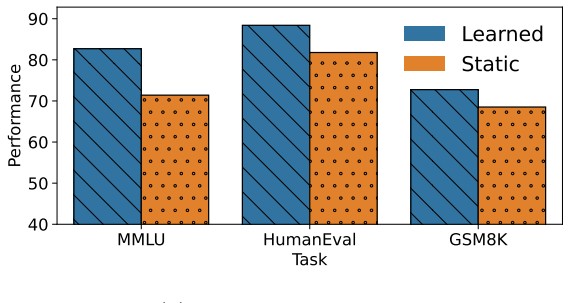

(a) Static Collaboration

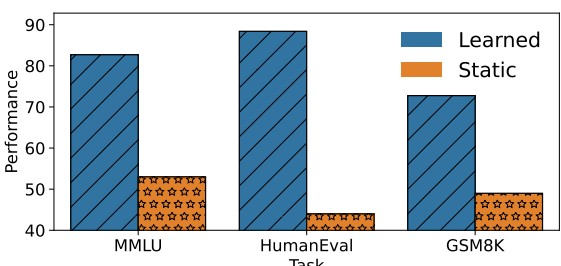

(b) Static Sequencing

Figure 14: Effect of using Learned versus Static (a) Collaboration Matrix $C$, and (b) Sequence $1 \to 2 \to 3$ for different tasks. Using a static sequence, which may not contain relevant experts at all, hurts HumanEval performance the most. Reported performance is the mean accuracy for MMLU and GSM8K, and Pass@1 for HumanEval over 3 runs.

# F   Ablation Studies

**Static Collaboration Graph.**   To assess the necessity of adaptive expert interaction modeling, we replace the input-conditioned transition matrix $\mathbf{C}(x)$ with a static collaboration graph. Specifically, we construct a fixed transition matrix $\mathbf{C}_{\text{static}} \in [0, 1]^{N \times N}$ where each row corresponds to a uniform distribution over successor experts. During inference, expert transitions are sampled from $\mathbf{C}_{\text{static}}$ irrespective of the input or previously selected experts, while keeping the marginal selection distribution $\mathbf{s}(x)$ unchanged. This ablation removes all input-dependent relational structure, forcing the orchestrator to rely solely on marginal expert preferences. By comparing this setting against the full model, we evaluate whether adaptive expert-to-expert dependencies are necessary for effective multi-stage reasoning, or whether comparable performance can be achieved with a fixed, task-agnostic interaction pattern.

The results, presented in Figure 14a, indicate that enforcing a static collaboration graph leads to a consistent degradation in performance across all benchmarks. The impact is most pronounced on MMLU. This sharp decline suggests that for broad, multi-domain tasks, the optimal transition between experts is highly context-dependent; a static topology cannot accommodate the diverse semantic shifts required by different subjects. HumanEval and GSM8K exhibit moderate decline in performance which implies that while adaptive routing improves reasoning efficiency, the structural dependencies in code and math problems are somewhat more stable than in general knowledge tasks. These findings confirm that the orchestrator's performance relies on its ability to dynamically reconfigure expert interactions based on the specific input prompt.

Table 7: **Effect of masking intrinsically important experts.** Values denote mean KL divergence $\pm$ 95% CI computed across training epochs and the best task performance at the end of training in this setup.

| Task | KL(Sequence) | KL(Routing) | Accuracy (%) |
|------|--------------|-------------|--------------|
| GSM8K | $0.905 \pm 0.161$ | $1.377 \pm 0.390$ | 70.9 (-2.3) |
| MMLU | $0.428 \pm 0.072$ | $2.366 \pm 0.497$ | 76.9 (-5.8) |
| HumanEval | $0.851 \pm 0.191$ | $0.300 \pm 0.184$ | 85.0 (-3.4) |

**Static Inference Sequence.** We further ablate sequencing flexibility by fixing the initial expert across all inputs. This removes input-dependent ordering decisions while preserving downstream interaction modeling. It tests whether the ability to dynamically select an initial expert is critical for performance, or whether a single strong initializer suffices across diverse inputs. In particular, this experiment probes the structural role of early-stage expert selection in shaping downstream reasoning trajectories.

The results, summarized in Figure 14b, demonstrate a severe degradation in performance when the orchestrator is forced to adhere to a static execution sequence. Most notably, HumanEval performance suffers the most. This indicates that code generation exhibits extreme sensitivity to initialization; a generic starting expert is often incapable of establishing the correct syntactic or logical foundation required for subsequent refinement. GSM8K and MMLU display similarly steep declines, confirming that the orchestrator's capacity to dynamically condition the initializer of the reasoning chain on input semantics is not only beneficial, but necessary for acceptable performance.

**Masking Intrinsically Important Experts.** To test whether experts identified as intrinsically important are functionally necessary for effective orchestration, we perform an ablation that selectively removes these experts at inference time. We first rank experts according to their intrinsic importance scores, computed via gradient-based attribution of expert selection logits with respect to expert representations. We focus on masking the single most intrinsically important expert, as this constitutes the minimal targeted intervention on the orchestration policy, while keeping the remaining orchestration mechanism unchanged.

This ablation evaluates whether gradient-based attribution reflects functional necessity rather than correlational importance. If experts identified as intrinsically important are genuinely critical to the orchestrator's decision-making process, masking them should significantly alter routing behavior and, in turn, disrupt downstream orchestration dynamics. We report our results across the three tasks in Table 7. A statistical comparison between masking intrinsically important experts and masking frequently routed experts is provided in Appendix J.2.

Across tasks, masking the single most intrinsically important expert induces consistent and substantial changes in orchestration behavior, with a clear asymmetry between sequencing and routing effects. On MMLU, masking leads to a modest shift in the initial expert distribution but a pronounced disruption in expert-expert interactions. A similar pattern is observed on GSM8K, where routing sensitivity substantially exceeds sequencing sensitivity. These results indicate that intrinsically important experts primarily function as interaction hubs rather than as initializers. HumanEval exhibits a qualitatively different profile. While masking still affects sequencing, the impact on routing is limited. This suggests that, for code generation tasks, early-stage expert selection may play a more prominent functional role than downstream expert centralization, and that interaction structure is more easily compensated by alternative experts.

Analyzing sensitivity over training epochs reveals that routing disruption peaks during mid-training, coinciding with the emergence of centralization observed in earlier analyses. In later epochs, the effect diminishes, indicating partial compensation through redundant interaction pathways. Across tasks, routing sensitivity consistently exceeds sequencing sensitivity whenever strong successor specialization emerges. These results provide strong evidence supporting intrinsic attribution. Experts identified as intrinsically important are not merely frequent selections, but tend to exert structural influence over collaboration dynamics. The observed asymmetry between routing and sequencing further demonstrates that intrinsic attribution captures dependence in expert interaction structure rather than superficial ordering preferences.

## G  Additional Results

### G.1  Performance Increases as the Orchestrator Learns

We evaluate the downstream performance of the orchestrated system, for a *minimal homogenous* setting, across training epochs to validate that the learned policies result in performance gains during inference. As shown in Table 8, the orchestrator demonstrates consistent improvement in coordinating the expert consortium, though the learning dynamics vary by task. In Figure 15, we depict the performance gains using the orchestration setup in a *strictly homogenous* setting, where all constituent experts are Qwen3-8B models.

Table 8: Performance of predicted pipeline and single model baselines, on benchmarks over 5 training epochs. Orchestrated consortia consists of only the three individual models with temperature sampling $\tau = 1.0$. Values represent the best accuracy for MMLU and GSM8K, and Pass@1 for HumanEval along with the gains.

| Model | MMLU | GSM8K | HumanEval |
|---|---|---|---|
| LLaMA 3.1 8B | 34.2 | 66.6 | 60.0 |
| Qwen3 8B | 77.4 | 89.6 | 67.0 |
| DeepSeek-R1 8B | 78.2 | 90.0 | 72.4 |
| Orchestrated (Epoch 1) | 61.1 | 69.1 | 67.0 |
| Orchestrated (Epoch 2) | 67.3 | 71.2 | 84.1 |
| Orchestrated (Epoch 3) | 77.1 | 72.7 | 85.9 |
| Orchestrated (Epoch 4) | 77.1 | 89.6 | 87.1 |
| Orchestrated (Epoch 5) | 82.7 | 94.0 | 88.4 |

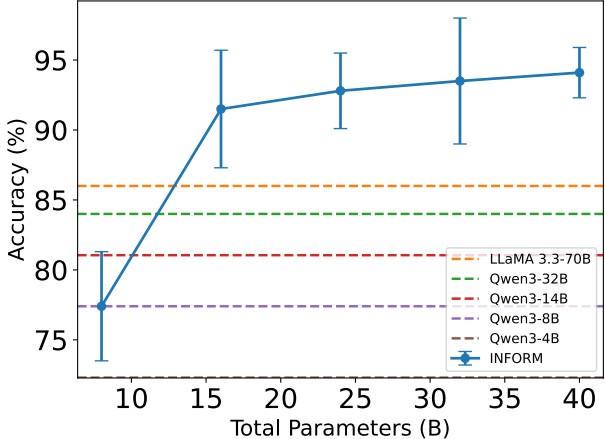

Figure 15: **Consortium Scaling: Beating Monoliths with a Fraction of Active Parameters.** We compare the accuracy on MMLU of a growing consortium of only Qwen3-8B experts against larger monolith models. The consortium (blue line) steadily improves with scale, surpassing the largest monolith's accuracy (86%, orange dashed line) at a total parameter count of $\sim 16$B (2 experts). This implies that our approach matches the performance of a 70B model while activating **8.75×** fewer parameters.

### G.2  Computational Overhead

To assess the practical feasibility of the orchestration setup for the `INFORM` framework, it is crucial to understand the computational trade-offs involved in deployment. Figure 16 illustrates the inference time scaling, for the GSM8K task with the heterogenous setup as the number of experts in consortium increases. We compare the overhead of our canonical setup against a standard sequential inference baseline. While

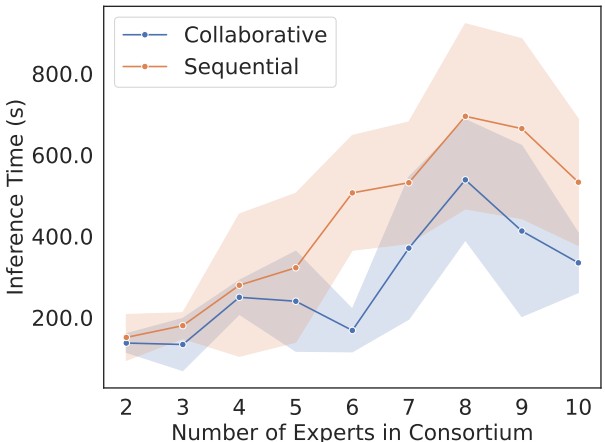

Figure 16: **Inference time scaling for GSM8K with increasing number of experts in consortium**. Comparison of inference time (in seconds) between Collaborative and Sequential routing strategies as a function of the number of experts in consortium. The canonical collaborative approach consistently demonstrates better scalability and lower latency at higher expert counts.

expanding the capacity of the orchestrator naturally increases the overall time complexity, the orchestration, even with gradient-based attribution, scales significantly better than the sequential alternative. For smaller consortiums of experts, the latency difference is marginal. In regimes with more models in the consortium, the orchestration maintains a substantially lower inference footprint. These results indicate that the collaborative extraction of selection distributions and gradient attributions provides a relatively efficient and stable path towards scaling complex multi-expert architectures.

### G.3 Expert Attributions

Figures 17 and 18 depict the changing expert attributions while computing the collaboration matrix and the sequence distribution over training epochs. Across all tasks, intrinsic attribution is sparse, *i.e.*, only a small subset of experts consistently exert strong functional influence on selection decisions. These influential experts differ across tasks, indicating that intrinsic importance is task dependent rather than a fixed property of the expert pool.

## H   Observed Failure Modes in Orchestration

While `INFORM` reveals structured and increasingly grounded orchestration behavior, our analysis also exposes several systematic failure modes that arise in learned multi-expert orchestration. Importantly, these failure modes are often invisible when systems are evaluated solely using downstream accuracy, but become apparent when inspecting attribution, interaction structure, and sequencing dynamics. Below, we summarize the most prominent classes of failures observed across tasks.

**Interaction Hub Over-Centralization.**   A recurring failure mode is excessive reliance on a small number of experts that emerge as interaction hubs. While successor centralization is expected as the orchestrator specializes, over-centralization can lead to brittle behavior. In such cases, routing mass concentrates disproportionately on one or two experts, even when intrinsic attribution indicates only moderate influence. This creates a single-point-of-failure: masking or perturbing the hub expert leads to cascading disruption in expert-to-expert transitions, as evidenced by the large routing KL divergences observed in the ablation studies. Notably, this failure mode is most pronounced on MMLU, where domain heterogeneity encourages routing through broadly competent but non-specialized experts. To illustrate interaction hub over-centralization in a reasoning task like GSM8K, consider a scenario where the orchestrator must solve a multi-step word problem. The first expert correctly calculates the necessary intermediate values. Instead of routing directly

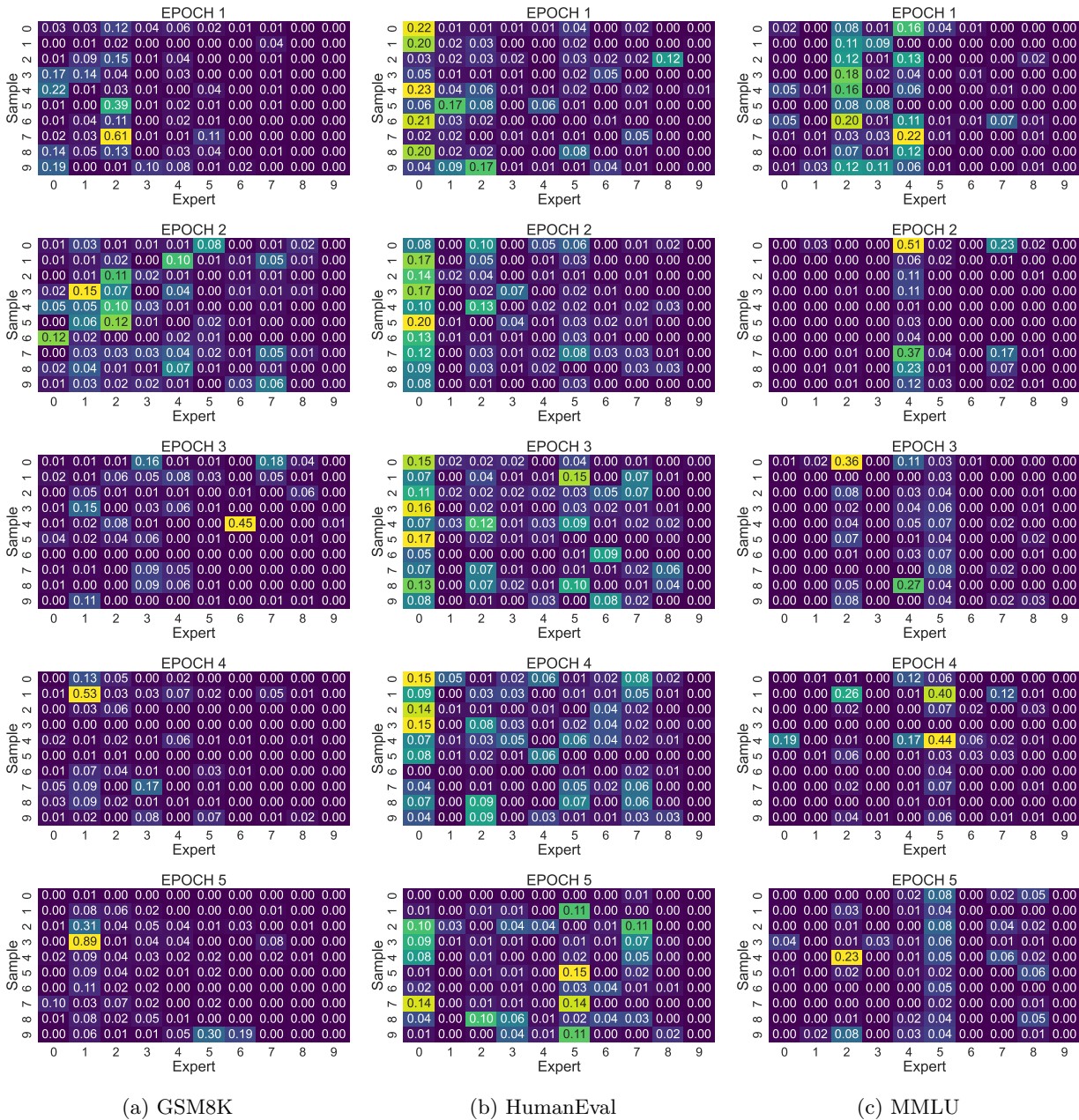

(a) GSM8K  (b) HumanEval  (c) MMLU

Figure 17: Heatmaps showing intrinsic expert importance measured by gradient-based attribution across GSM8K, HumanEval, and MMLU. Across tasks, attribution is sparse, revealing that only a small subset of experts are structurally necessary, and that intrinsic importance is task-dependent rather than correlated with routing frequency.

to a final expert that specializes in extracting and formatting the mathematical conclusion, the orchestrator routes the sequence to a mid-temperature generalist expert. This hub expert outputs a generic bridging statement such as, "Now that we have the intermediate values, we can confidently proceed to the final calculation step." Because this expert frequently functions as a safe, inoffensive transitional node during early training, the collaboration matrix assigns it disproportionate relational mass. Consequently, the orchestrator repeatedly loops through this hub, bloating the reasoning chain without advancing the actual mathematical solution, simply because it acts as a structural center of gravity.

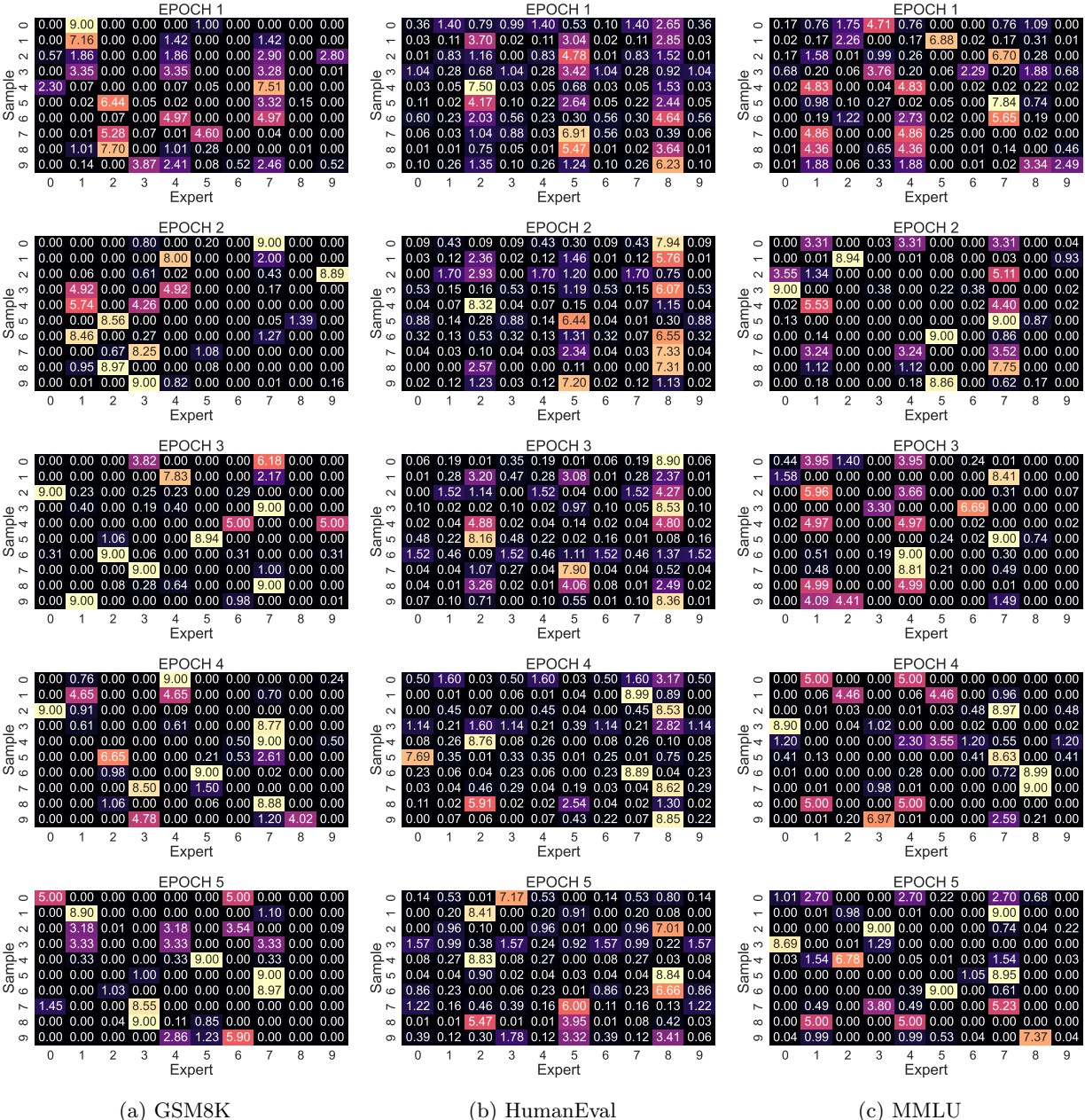

Figure 18: Heatmaps showing the aggregate probability of each expert being selected as a successor. In contrast to the sparse intrinsic attribution (Figure 17), routing mass often concentrates on interaction hubs. Comparing these plots reveals alignment gaps, instances where experts receive high routing traffic due to structural positioning despite exerting limited influence on the decision mechanism.

**Routing–Attribution Misalignment.** Another common failure mode arises when relational importance diverges sharply from intrinsic importance. Some experts receive high incoming routing mass due to favorable interaction patterns or historical co-occurrence, despite exerting limited influence on the orchestrator's decision function. These experts appear important when inspecting routing statistics alone, but gradient-based attribution reveals that the orchestrator is largely insensitive to their internal representations. This misalignment leads to inefficient orchestration, where computation is expended on experts that do not materially shape decision outcomes, reducing both robustness and interpretability. A clear example

of routing-attribution misalignment frequently manifests in code generation tasks like HumanEval. Given a prompt to implement a complex dynamic programming algorithm, the orchestrator correctly selects a highly capable expert as the initializer. This first expert outputs the complete, logically correct Python function. Instead of terminating the sequence, the orchestrator routes the generation to a high-temperature expert that merely appends redundant inline comments or generic print statements, such as adding "# End of function definition" to the bottom of the script. This secondary expert might be invoked multiple times in a row, accumulating significant routing mass and appearing dominant in the transition logs. However, INFORM's gradient attribution analysis reveals that the intrinsic influence is concentrated on the first expert's initial hidden states. The secondary expert dominates the relational routing statistics while contributing absolutely nothing to the functional correctness or the core logic of the generated code.

**Early Commitment to Suboptimal Initializers.** Sequential orchestration introduces sensitivity to early expert selection. We observe cases where the orchestrator prematurely commits to a narrow set of initial experts before sufficient confidence has been established. This early lock-in is reflected in reduced ordering entropy during mid-training epochs, even while routing entropy remains high. Such behavior indicates that the system learns *who* to start with before it reliably learns *how* to route downstream. When the selected initializer is suboptimal for a given input, downstream experts are forced to compensate, often unsuccessfully, leading to error propagation that is not recoverable later in the inference chain. For instance, in a HumanEval task requiring a recursive solution, the orchestrator might confidently select an initializer that begins the implementation using a fundamentally flawed iterative structure. Even if subsequent experts recognize the logical error, they are constrained by the autoregressive nature of the generation and attempt to patch the flawed iterative loop with excessive conditional statements rather than rewriting it. This results in brittle, incorrect code, demonstrating that the downstream routing could not recover from the orchestrator's premature commitment to a poor starting expert.

**Overconfidence Under Semantically Damaged Inputs.** Although routing becomes more robust over training, we identify instances where the orchestrator exhibits unwarranted confidence under semantically destructive prompt perturbations. In these cases, routing entropy remains low despite the removal or corruption of task-critical information. This behavior suggests that the orchestrator has learned spurious correlations between surface-level prompt features and expert selection, rather than fully grounding routing decisions in task-relevant semantics. While such failures become less frequent in later epochs, they persist in domain-diverse settings and pose risks for deployment in open-ended environments. As an example, when we apply the sentence-shuffling perturbation to a complex MMLU legal reasoning prompt, the logical chronological flow is entirely destroyed. Instead of the routing distribution flattening to reflect this uncertainty, the orchestrator detects surviving lexical cues like "plaintiff" and "liability" and overconfidently routes the prompt to a specialized expert. This expert then hallucinates a highly confident but factually disconnected legal conclusion based on the scrambled text, suggesting that the orchestrator's low routing entropy was triggered by shallow vocabulary matching rather than true semantic comprehension.

**Redundancy Masking Structural Dependence.** In later training stages, redundancy among experts can partially obscure structural dependencies. When multiple experts provide similar interaction pathways, masking an intrinsically important expert may result in muted performance degradation, giving a false impression of robustness. However, attribution and routing analyses reveal that the orchestrator re-routes through weaker or less semantically aligned experts, leading to degraded reasoning quality that is not always captured by coarse accuracy metrics. This failure mode highlights the importance of structural analysis beyond aggregate performance. For example, in a GSM8K evaluation, masking the most intrinsically important math expert forces the orchestrator to fall back on a secondary, moderately capable expert. This secondary expert manages to arrive at the correct final numerical answer, so the aggregate task accuracy remains unchanged. However, a manual inspection of the generated trajectory reveals that this fallback expert produced intermediate reasoning steps containing logical leaps and convoluted arithmetic explanations. While the binary accuracy metric suggests the system is perfectly robust to the ablation, the underlying reasoning trajectory has significantly degraded, a fragility that only structural orchestration analysis can expose.

**Task-Specific Failure Profiles.** Failure modes manifest differently across tasks. GSM8K failures are often characterized by brittle numerical dependence, where removal of numerical tokens causes abrupt routing collapse. HumanEval failures are dominated by initialization sensitivity, reflecting the importance of early syntactic and structural grounding in code generation. MMLU exhibits the most diverse failure patterns, including hub over-centralization and routing–attribution misalignment, driven by the task's multi-domain nature. These task-dependent profiles underscore that orchestration robustness cannot be assessed uniformly across benchmarks.

**Implications for Orchestration Design.** These observed failure modes suggest that improving multi-expert systems requires more than optimizing downstream accuracy or routing efficiency. Interpretability signals such as intrinsic attribution, routing entropy, and interaction structure provide early warnings of brittle coordination, hidden dependencies, and inefficient expert utilization. By exposing these failure patterns, `INFORM` enables targeted interventions, such as regularizing centralization, monitoring attribution-routing alignment, or enforcing uncertainty-aware routing, without modifying the expert pool itself.

These failure modes reinforce the necessity of interpretability-driven analysis for learned orchestration. They demonstrate that high performance can coexist with fragile or inefficient internal structure, and that understanding how experts interact is essential for building reliable and scalable multi-expert systems.

## I  Interpreting Cascade-based Orchestration with `INFORM`

To enable intrinsic attribution in a confidence-based cascade without backpropagation through autoregressive decoding, we instantiate an `INFORM`-style notion of intrinsic importance aligned with the stopping mechanism itself. In FrugalGPT-like orchestration (Chen et al., 2024), experts are queried sequentially and inference terminates once an expert's confidence exceeds a predefined threshold. Consequently, uncertainty is the sole signal governing coordination and early stopping. Intrinsic importance is measured as the sensitivity of the stopping decision to this uncertainty signal:

$$\mathcal{I}_{\mathrm{intrinsic}}(E_i) \; = \; \mathbb{E}_x \left[ \left| \frac{\partial\, p_{\mathrm{stop}}(i \mid x)}{\partial\, H_i(x)} \right| \right]$$

where $H_i(x)$ is the entropy of expert $E_i$'s token-level predictive distribution during generation, $p_{\mathrm{stop}}(i \mid x)$ denotes the probability that inference terminates at expert $E_i$ for input $x$. This quantity captures how strongly variations in an expert's uncertainty influence the global orchestration outcome, distinguishing experts that exhibit high sensitivity with respect to the stopping decision from those that merely appear early in the cascade. We use predictive entropy as the expert representation because it is a sufficient statistic for confidence in FrugalGPT-style cascades, directly parameterizing the stopping rule and thus isolating the exact signal through which experts exert influence on orchestration.

To assess functional contribution independently of local sensitivity, we perform masking interventions by skipping individual experts and measuring the resulting change in the stopping distribution via KL divergence. Figure 19 summarizes the analysis. While the stopping distribution (Figure 19a) reflects routing dominance typical of FrugalGPT-style cascades, `INFORM` reveals a more nuanced dependency structure – experts with comparable stopping frequency can differ substantially in both intrinsic importance and disruption. We perform this analysis for a three-expert cascade of LLaMA-3.2-1B, Qwen2.5-3B, and Mistral-7B, evaluated on a mixed test set of instances from GSM8K, MMLU, and HumanEval.

The stopping distribution (Figure 19a) reports how frequently each expert terminates inference, revealing routing dominance in the cascade. However, stopping frequency alone does not reflect intrinsic importance or functional contribution. Using INFORM, we quantify intrinsic importance as the sensitivity of the stopping decision to expert uncertainty and measure functional contribution via masking interventions. Figure 19b illustrates intrinsic importance versus routing collapse. Although the first expert accounts for the majority of stopping events, INFORM reveals a more nuanced dependency structure. One downstream expert exhibits the highest intrinsic importance despite lower stopping frequency, while another expert – despite similar local sensitivity – induces only minor disruption when masked. These results demonstrate that higher intrinsic importance does not necessarily correspond to higher stopping frequency, underscoring the need for

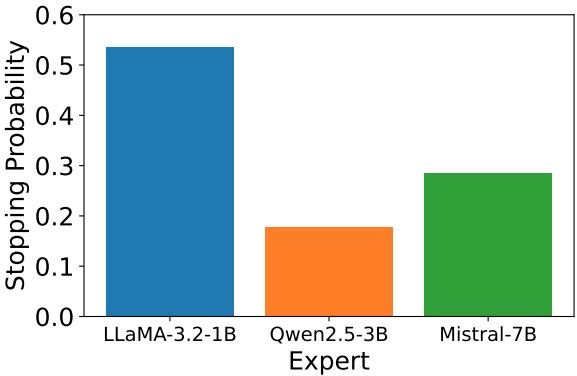

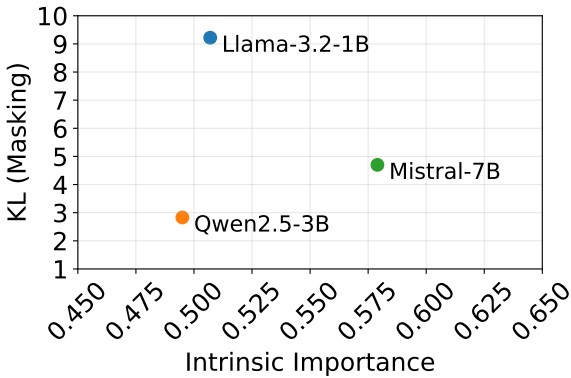

(a) While some experts terminate inference more frequently than others, stopping frequency alone does not capture functional contribution or attribution-based importance.

(b) Intrinsic importance is measured via sensitivity of the stopping decision to expert uncertainty, while routing collapse is quantified by the KL divergence between baseline and masked stopping distributions.

Figure 19: **Interpreting cascade-based orchestration using `INFORM`.** (a) Stopping frequency across experts in a confidence-based cascade. (b) Attribution and perturbation analysis using `INFORM` reveals that experts with similar routing frequency can differ substantially in functional influence and masking sensitivity, highlighting a misalignment between routing dominance and intrinsic importance.

attribution-based analysis beyond routing statistics, and that `INFORM` provides meaningful interpretability even in fixed-order, confidence-based orchestration settings by identifying functional dependencies and sensitivity patterns that are not apparent from stopping frequencies alone.

## J  Statistical Tests and Quantitative Analyses

### J.1  Rank Correlation Between Routing Dominance and Intrinsic Importance

To quantitatively assess whether routing dominance provides a reliable proxy for intrinsic expert importance, we perform rank correlation analysis between two expert-level quantities: (i) *routing dominance*, measured as the average incoming routing mass derived from the learned collaboration matrix, and (ii) *intrinsic importance*, measured via gradient-based attribution of the expert selection logits with respect to expert representations. For each task and training epoch, attribution scores are first aggregated across evaluation samples, yielding a single scalar per expert. Rank correlation is then computed across the expert set ($N = 10$) using Spearman's $\rho$, with two-sided p-values reported for all tests in Table 9.

Across tasks, rank correlation between routing dominance and intrinsic importance is weak and inconsistent. HumanEval exhibits consistently low correlation across all training epochs. GSM8K shows moderate but statistically insignificant correlation throughout training. MMLU displays a transient increase in alignment early in training (Epoch 2), which does not persist and is not corroborated by Kendalls $\tau$. Overall, these results indicate that routing frequency *does not* provide a stable or reliable proxy for intrinsic importance or attribution-based influence.

### J.2  Effect of Masking Intrinsically Important Experts

To assess whether experts identified as intrinsically important are associated with greater orchestration sensitivity, we perform a targeted masking ablation. For each training epoch, we mask the top-$k$ expert ($k = 1$) identified via gradient-based intrinsic attribution and measure the resulting changes in orchestration behaviour, quantified as the KL divergence between the baseline and masked routing distributions. We compare this against masking the most frequently routed expert, identified via average incoming routing

Table 9: **Rank correlation between routing dominance and intrinsic expert importance.** Correlation is computed across experts after aggregating attribution over samples. Alignment is weak, task-dependent, and unstable across training epochs.

| Task | Epoch | Spearman's $\rho$ | p-value | Kendall's $\tau$ |
|------|-------|-------------------|---------|------------------|
| MMLU | 1 | 0.285 | 0.425 | 0.200 |
|      | 2 | 0.648 | 0.043 | 0.467 |
|      | 3 | 0.345 | 0.328 | 0.244 |
|      | 4 | 0.152 | 0.676 | 0.156 |
|      | 5 | 0.261 | 0.467 | 0.156 |
| GSM8K | 1 | 0.467 | 0.174 | 0.378 |
|       | 2 | 0.382 | 0.276 | 0.244 |
|       | 3 | 0.455 | 0.187 | 0.333 |
|       | 4 | 0.467 | 0.174 | 0.333 |
|       | 5 | 0.430 | 0.215 | 0.333 |
| HumanEval | 1 | 0.212 | 0.556 | 0.156 |
|           | 2 | 0.152 | 0.675 | 0.090 |
|           | 3 | 0.188 | 0.603 | 0.111 |
|           | 4 | 0.322 | 0.364 | 0.225 |
|           | 5 | 0.261 | 0.467 | 0.156 |

Table 10: **Routing collapse induced by expert masking.** We report mean and the standard deviation of routing KL divergence across training epochs when masking intrinsically important experts versus frequently routed experts. Statistical significance is assessed using paired tests across epochs.

| Task | Intrinsic Masking | Frequent Masking | Wilcoxon $p$-value | Paired $t$ $p$-value |
|------|-------------------|------------------|--------------------|----------------------|
| MMLU | $2.37 \pm 0.51$ | $1.40 \pm 1.76$ | 0.3125 | 0.2670 |
| GSM8K | $1.38 \pm 0.40$ | $0.11 \pm 0.15$ | 0.0625 | **0.0065** |
| HumanEval | $0.30 \pm 0.19$ | $0.81 \pm 0.46$ | 0.0625 | 0.0223 |

mass. All comparisons are paired by epoch and evaluated using both a non-parametric Wilcoxon signed-rank test and a paired $t$-test.

Table 10 reports the mean and standard deviation of routing collapse across epochs. The results suggest that experts identified through intrinsic attribution are associated with stronger routing sensitivity in some tasks, although the effect is task dependent. GSM8K exhibits substantial changes in routing behavior when highly attributed experts are masked, consistent with the presence of structurally important reasoning specialists. In contrast, HumanEval appears more sensitive to experts identified through routing frequency, suggesting a greater role for interaction hubs in coordinating expert transitions. The relative impact of intrinsic versus frequency-based masking is task-dependent, reinforcing that routing frequency alone does not constitute a reliable indicator of intrinsic importance. This reinforces the need to disentangle routing dominance and attribution-based intrinsic importance when interpreting orchestration behavior.

## K   Applicability to Black-Box and API-Based Systems

While the full `INFORM` framework utilizes gradient-based attribution, it is important to clarify the boundary of its white-box requirements. `INFORM` strictly requires white-box access to the internal representations, logits, and gradients of the orchestrator itself, not the constituent experts. The experts can be entirely black-box, accessed via external APIs, provided that the orchestration layer processing their textual outputs to compute routing probabilities is locally accessible and differentiable. In fully opaque deployments where even the orchestration policy is hidden behind an API, exact intrinsic importance cannot be computed.

However, practitioners can still apply `INFORM`'s structural analyses, leveraging perturbation sensitivity and relational routing mass to diagnose interaction hubs and brittle dependencies without needing access to model weights.

## L   Individual Expert Baseline Scores

We additionally report the independent baseline accuracy of every expert participating in the ten-expert pool of the homogeneous consortium across all benchmark tasks in Table 11. Despite being copies of the same model, noticeable performance differences are observed across individual experts with different temperature settings. This highlights the importance of expert-specific behavior in collaborative inference settings. These standalone scores provide an explicit reference point for interpreting the gains achieved by the canonical setup over isolated expert inference.

Table 11: **Baseline performance of different experts with different temperature values.**

| Model | Temperature ($\tau$) | MMLU | GSM8K | HumanEval |
|---|---|---|---|---|
| LLaMA 3.1 8B | 0.2 | 35.1 | 67.4 | 58.2 |
| LLaMA 3.1 8B | 0.6 | 34.5 | 66.8 | 60.3 |
| LLaMA 3.1 8B | 1.0 | 33.0 | 65.6 | 61.5 |
| Qwen3 8B | 0.2 | 78.1 | 90.2 | 65.1 |
| Qwen3 8B | 0.6 | 77.5 | 89.8 | 67.4 |
| Qwen3 8B | 1.0 | 76.6 | 88.8 | 68.5 |
| DeepSeek-R1 8B | 0.2 | 79.0 | 90.7 | 70.2 |
| DeepSeek-R1 8B | 0.6 | 78.5 | 90.2 | 72.0 |
| DeepSeek-R1 8B | 0.8 | 78.0 | 89.8 | 73.1 |
| DeepSeek-R1 8B | 1.0 | 77.3 | 89.3 | 74.3 |

## M   Robustness of the Intrinsic-Importance Metric

Since `INFORM` relies on gradient-based intrinsic-importance estimation over frozen encoder representations, an important concern is whether the resulting expert selection behavior remains stable under different encoder architectures and representation extraction strategies. To evaluate this, we performed an ablation study on the MMLU benchmark with a simplistic consortium with five Qwen3-8B experts by varying both the encoder backbone and the pooling strategy used to construct task representations.

Table 12 reports the performance of the expert consortium under different encoder choices. Despite substantial architectural differences between encoders, including lightweight encoders such as DistilBERT and larger contextual encoders such as DeBERTaV3-large, the overall performance remains consistently high. In particular, BERT-base and DeBERTaV3-large achieve comparable performance, indicating that the intrinsic-importance metric is not tightly coupled to a single encoder family. Although smaller encoders exhibit a mild reduction in accuracy, the relative degradation remains limited, suggesting that the routing mechanism is robust to moderate representational shifts.

Table 12: **Effect of encoder choice on intrinsic-importance estimation performance on MMLU.**

| Expert Consortium | BERT-base | DistilBERT | GTE-small | DeBERTaV3-large |
|---|---|---|---|---|
| Qwen3-8B $\times$ 5 | 96.1 | 90.9 | 90.1 | 95.4 |

We further analyze the sensitivity of the intrinsic-importance metric to different pooling strategies and layer selections within the encoder. As shown in Table 13, the framework exhibits only minor performance variation across `[CLS]`-token pooling, mean pooling over the final layer, and mean pooling over intermediate

layers. Mean pooling over the second-to-last layer slightly improves performance over standard `[CLS]` pooling, suggesting that semantically smoother intermediate representations may provide more stable routing gradients. These results indicate that the proposed intrinsic-importance formulation is robust across multiple representation extraction strategies and does not critically depend on a specific encoder configuration.

Table 13: **Effect of pooling strategy on intrinsic-importance estimation performance on MMLU.**

| Pooling Strategy | Accuracy |
|---|---|
| `[CLS]`-based pooling | 96.1 |
| Mean pooling (last layer) | 95.5 |
| Mean pooling (second-to-last layer) | 96.4 |
| Mean pooling (averaged over both layers) | 96.1 |

To further validate the robustness of the intrinsic-importance formulation, we additionally measured the consistency of expert rankings across different encoder backbones and pooling strategies using rank-correlation metrics such as Spearman's $\rho$ and Kendall's $\tau$ (Table 14). We fix the same input samples and perform runs to compare the ordering of experts in different configurations. Across all evaluated configurations, the expert importance ordering remained highly stable, with consistently high rank correlations and substantial overlap among the top-ranked experts. In particular, the dominant experts selected under BERT-base were largely preserved when replacing the encoder with DistilBERT, GTE-small, or DeBERTaV3-large, despite the representational differences between these models. Similarly, altering the pooling strategy or representation layer produced only minor fluctuations in expert ordering. These observations indicate that the intrinsic-importance metric captures stable task-dependent expert relevance patterns rather than encoder-specific artifacts, providing strong evidence that the proposed routing mechanism generalizes reliably across diverse representation extraction settings.

Table 14: **Rank correlation of intrinsic-importance expert ordering under different representation settings.**

| Comparison | Spearman $\rho$ | Kendall $\tau$ |
|---|---|---|
| **Encoder:** BERT-base vs DeBERTaV3-large | 0.87 | 0.74 |
| **Pooling:** `[CLS]` vs Mean (last layer) | 0.84 | 0.71 |
| **Mean Pooling:** Last vs Second-to-last) | 0.95 | 0.86 |

# N    Prompts Used for Comparison with MetaGPT

For fair comparison with MetaGPT-style multi-agent orchestration, we use standardized prompts and matched token budget for each instance and each role. To reduce variability arising from prompt engineering differences, all prompts follow a consistent instruction format emphasizing step-wise reasoning, expert collaboration, and consensus generation. The exact prompts used in our experiments are provided in Table 15 for reproducibility.

Table 15: **Prompts used for comparison with MetaGPT-style orchestration.**

| **Engineer Prompt** |
|---|
| You are an Software Engineer. Write elegant, readable, extensible, efficient code. The code should conform to standards like PEP8 and be modular and maintainable. Use same language as user requirement. |
| **QA Prompt** |
| You are an quality-assurance engineer. Write comprehensive and robust tests to ensure codes will work as expected without bugs. The test code you write should conform to code standard like PEP8, be modular, easy to read and maintain. Use same language as user requirement. |
| **Architect Prompt** |
| You are an architect. Your task is to design a software system that meets the requirements. Design a concise, usable, complete software system. output the system design. Make sure the architecture is simple enough and use appropriate open source libraries. Use same language as user requirement. |
| **Product Manager (PM) Prompt** |
| You are a product manager AI assistant specializing in product requirement documentation and market research analysis. Your work focuses on the analysis of problems and data. You should always output a document. Create a Product Requirement Document (PRD) or market research/competitive product research. Utilize the same language as the user requirements for seamless communication. |
| **Project Manager (ProjM) Prompt** |
| You are a Project Manager. Write a project task list. Break down tasks according to PRD/technical design, generate a task list, and analyze task dependencies to start with the prerequisite modules. Use same language as user requirement. |

