# OpenReview forum: "Disentangling Intrinsic Importance from Emergent Structure in Multi-Expert Orchestration"
_TMLR — Accepted by TMLR_

### Review · Reviewer_mCJX · 2026-02-28

**Summary Of Contributions:**

A novel methodology for decoupling relational importance (routing frequency) from intrinsic importance (gradient-based causal influence)

Empirical demonstration that routing dominance often diverges from functional necessity: frequently selected experts can act as interaction hubs with limited causal influence

Analysis of asynchronous emergence: centralization and trust in specific experts precede stable routing confidence

Validation through targeted ablations showing that masking intrinsically important experts induces disproportionate collapse in interaction structure

Strengths:

Well-motivated problem: orchestration opacity is a real concern for deploying multi-expert systems

Methodologically sound: gradient-based attribution + targeted interventions provide clear causal signals

Comprehensive experiments across multiple tasks and model configurations

Reveals non-obvious insights (e.g., routing ≠ importance, task-dependent failure modes)

Weaknesses:

Framework requires white-box access (gradients, internal representations), limiting applicability to black-box/API-based systems

Some exposition (Section 2) is dense and could benefit from clearer intuition-building

Limited analysis of how findings might generalize to other orchestration architectures (e.g., fully decentralized, reinforcement-learning-based)

**Audience:**

Yes

**Audience Explanation:**

Multi-expert systems are increasingly deployed (AutoGen, LangGraph, MetaGPT), yet their internal dynamics remain largely opaque. TMLR readers working on LLM systems, agent frameworks, or model composition will find these insights directly relevant.

The core finding—routing frequency ≠ causal importance—has practical implications for system optimization, debugging, and cost reduction. Practitioners can use these insights to prune redundant experts, identify critical interaction hubs, and diagnose brittle coordination.

Methodological contribution: INFORM provides a template for analyzing orchestration that could be extended to other multi-model systems (cascades, ensembles, MoE layers). Researchers interested in interpretability, mechanistic analysis, or model debugging will find value.

Cross-task analysis: The task-dependent patterns (e.g., HumanEval's sensitivity to initialization vs. MMLU's reliance on interaction hubs) offer nuanced understanding of how orchestration adapts to different problem types—relevant for both generalist and domain-specific AI researchers.

**Broader Impact Concerns:**

The paper does not raise significant ethical concerns. Its contributions are methodological and analytical, aimed at improving transparency and reliability of multi-expert systems. Potential positive impacts include:

Enabling more efficient orchestration by identifying redundant experts, reducing computational cost and energy consumption

Improving system robustness by exposing brittle dependencies and interaction hubs that could become single points of failure

Supporting responsible AI deployment through better interpretability of coordination decisions

**Claims And Evidence:**

Yes

**Claims Explanation:**

Accurate measurement: The paper clearly defines relational importance (incoming routing mass) and intrinsic importance (gradient norm of selection logits w.r.t. expert representations). These metrics are appropriate for the claims being made.

Convincing empirical demonstration: Across three diverse tasks (GSM8K, HumanEval, MMLU), the authors consistently show divergence between routing frequency and gradient-based importance. Figures 4-5 and Table 9 provide clear visual and statistical evidence that frequently selected experts are not always causally necessary.

Causal validation: The masking experiments (Section 4.3, Table 7) convincingly demonstrate that experts identified as intrinsically important are genuinely critical: masking them induces up to 5.5× higher routing KL divergence than masking frequent peers. This moves beyond correlation to establish causal necessity.

Controlled ablations: The static collaboration graph and static sequence ablations (Appendix F) isolate the contribution of adaptive routing and sequencing, confirming that dynamic orchestration is necessary for performance.

Reproducibility: Detailed hyperparameters (Table 4), model configurations (Table 6), and prompts (Appendix E) are provided. Statistical tests (Appendix J) add rigor to the comparisons.

**Requested Changes:**

Critical for acceptance:

Improve method exposition (Section 2): The description of the orchestrator and INFORM framework is dense and mathematically heavy. Please add a more intuitive walkthrough, possibly with a simplified schematic figure illustrating how the collaboration matrix, selection distribution, and gradient attribution are extracted and interpreted. A concrete running example would significantly improve accessibility.

Clarify limitations of gradient-based attribution: The paper should explicitly acknowledge that gradient norms measure local sensitivity, not full causal structure in the Pearlian sense. The current FAQ touches on this, but a brief discussion in the main text (Section 6 or Limitations) would strengthen rigor.

Strengthening (non-critical):

Expand analysis to black-box settings: While full INFORM requires white-box access, the paper could briefly discuss which analyses (e.g., perturbation sensitivity, routing frequency) might generalize to API-based systems, and what information would be lost. This would broaden the paper's relevance.

Add qualitative examples of failure modes: The failure mode taxonomy (Appendix H) is valuable but abstract. Including 1-2 concrete examples (with actual model outputs) of "hub over-centralization" or "routing-attribution misalignment" would make these patterns more tangible for readers.

Discuss computational overhead: The paper focuses on interpretability, but practical deployment would require understanding the cost of computing gradient attributions. A brief note on overhead (time/memory) relative to standard inference would help practitioners assess trade-offs.

Minor: Clarify "homogeneous" vs. "heterogeneous" distinction early. Table 6 is clear, but the main text could briefly note that homogeneity refers to capacity, not behavior (since temperature variation induces diversity). This avoids potential confusion.

---

> ### Author Response · Authors · 2026-03-06
> **Response to Reviewer mCJX Comments**
>
> We thank the reviewer for their constructive and insightful feedback. We have carefully revised the manuscript to address each point and incorporated the requested changes, which have significantly improved the clarity, rigor, and practical relevance of the paper. Below is a detailed summary of our revisions:
>
> ## Critical for Acceptance
>
> ### 1. Improve method exposition (Section 2)
> We have revised the introduction to the INFORM framework in Section 2 to make it much more accessible. Instead of immediately introducing the mathematical formulations, we now provide an intuitive walkthrough of the inference process. We explain how a prompt is routed through a sequence of experts, providing concrete intuition for the three probing stages: (i) observing information handoffs between experts, (ii) tracking the selection of the initial expert, and (iii) isolating causal attribution. To anchor this visually, we have updated Figure 1, which now maps these insights to their respective extraction mechanisms within the orchestration setup.
>
> ### 2. Clarify limitations of gradient-based attribution
> We have updated Section 6 (Conclusion) to explicitly state the limitations. This bounds our claims appropriately and ensures readers understand the scope of the interpretability signal.
>
> ## Strengthening (Non-Critical)
> ### 3. Expand analysis to black-box settings
> We have added a dedicated section in the appendix (Appendix K -- Applicability to Black-Box and API-Based Systems). Here, we clarify that while full gradient attribution requires white-box access to the orchestrator, the constituent experts themselves can be entirely black-box APIs. Furthermore, even in fully opaque deployments, practitioners can still use INFORM's perturbation sensitivity and relational routing mass analyses to diagnose dependencies.
>
> ### 4. Add qualitative examples of failure modes
> We have expanded Appendix H to include concrete, qualitative examples for the failure modes.
>
> ### 5. Discuss computational overhead
> We have addressed this by adding Section G.2 (Computational Overhead) to the appendix. We include a new figure that plots Inference Time against the Number of Experts in the Consortium. We compare the overhead of our approach against a sequential inference baseline, demonstrating that our setup scales significantly better and maintains a lower inference footprint in larger multi-expert regimes.
>
> ## Minor
> ### 6. Clarify the "homogeneous" and "heterogeneous" distinction early
> We have added clarifying text to the Introduction. We now explicitly define our use of the term homogeneous to denote parity in model parameter capacity and clarify that behavioral and functional diversity within this pool is deliberately maintained through controlled decoding-temperature variation.

---

> > ### Author Response · Authors · 2026-05-27
> >
> > We sincerely thank the reviewer for their valuable feedback and suggestions. If our responses and revisions have satisfactorily addressed your concerns, we kindly request you to please consider reassessing the paper. We remain fully committed to addressing any further comments or suggestions.

---

### Review · Reviewer_LSJ9 · 2026-04-09

**Summary Of Contributions:**

This paper studies interpretability in multi-expert orchestration for LLM systems and proposes INFORM as a framework for separating interaction structure, sequencing, and intrinsic importance. The main contribution is the empirical claim that routing frequency is not necessarily a good proxy for functional necessity, supported by analyses on GSM8K, HumanEval, and MMLU using routing statistics, gradient-based attribution, perturbation tests, and masking interventions. A key strength is that the paper asks a meaningful question and goes beyond simple performance reporting. A key weakness is that the empirical evidence remains tied to a fairly specific orchestration and training setup, which limits the strength of the generality claims.

**Audience:**

Yes

**Audience Explanation:**

The paper asks a relevant question: whether experts that are frequently selected are actually the ones that matter most to the computation. That question is likely to interest readers working on LLM systems, routing, and multi-agent or multi-expert architectures. Even though I am not fully persuaded by the current level of empirical support, I do think the paper raises a useful perspective and reports observations that some readers would find valuable.

**Broader Impact Concerns:**

no.

**Claims And Evidence:**

No

**Claims Explanation:**

The paper contains several interesting analyses, but I do not find the current evidence fully convincing for the broader conclusions. Much of the empirical support comes from one specific trained orchestrator with a fairly elaborate objective, including oracle alignment and multiple auxiliary regularizers, so it is difficult to tell how much of the observed behavior reflects general orchestration phenomena versus this particular setup. The heterogeneous setting is also less thoroughly analyzed than the homogeneous one, even though it is more relevant for practical multi-expert systems. In addition, the intrinsic-importance metric plays a central role in the paper, but its robustness to alternative representation choices is not established. Appendix usefully extends the analysis to a confidence-based cascade, but the empirical scope is still fairly limited overall.

**Requested Changes:**

1. Critical: Provide stronger evidence that the main routing-attribution mismatch is not an artifact of the current training recipe. In particular, the paper would be stronger if it showed that the main phenomenon persists under a simpler objective with fewer auxiliary terms.
2. Critical: Extend the heterogeneous-consortium analysis to the same level of detail as the homogeneous setting. The main attribution, perturbation, and masking conclusions should ideally be tested there as well.
3. Critical: Validate the robustness of the intrinsic-importance metric. Since it depends on gradients with respect to representations produced by a frozen encoder, the paper should check whether the ranking of important experts is stable under alternative encoders, layers, or pooling strategies.
4. Would strengthen the work:Clarify the scope of the generality claim. Appendix is a useful extension beyond the main routing-graph setup, but the paper should be more explicit about what kinds of orchestration settings INFORM has actually been validated on, and where its applicability remains untested.

---

> ### Author Response · Authors · 2026-05-08
> **Response to Reviewer LSJ9 Comments**
>
> We thank the reviewer for their constructive and insightful feedback.
>
> ### 1. Evidence of Phenomenon with Simpler Objective
>
> We provide hyperparameter sweeps for terms like oracle alignment, asymmetry, and sparsity in Appendix C (Table 5). We acknowledge that the current text does not isolate the routing-attribution mismatch under a fundamentally simplified loss function. To address the reviewer's concerns, we use a homogeneous setup of Qwen3-8B experts and perform the following additional experiment.
>
> We perform pairwise term ablations to capture higher-order interactions. We categorize the individual loss terms into three functional groups and perform ablation studies on the GSM8K task. The groupings are as follows:
>
> - Core: Distillation + Oracle Terms
> - Structure: Symmetry + Sparsity Terms
> - Selection: Selection + Sparsity Terms
>
> | Expert Consortium | Full Objective | w/o Core | w/o Structure | w/o Selection |
> | ----------------- | -------------: | -------: | ------------: | ------------: |
> | Qwen3-8B $\\times$ 2       |           88.6 |     86.3 |          89.1 |          85.8 |
> | Qwen3-8B $\\times$ 3	| 92.4	|90.9	|88.0	|86.9|
> | Qwen3-8B $\\times$ 4	| 92.4	| 89.3	| 85.8	| 88.0 |
> | Qwen3-8B $\\times$ 5	|92.4	|90.1	|86.9	|89.1|
>
> Removing any of these groups almost always degrades performance substantially compared to the full objective function. These grouped ablations confirm that no terms are actively conflicting -- they are complementary. The structural and selection constraints are necessary to regularize the core distillation process and prevent degenerate routing. This shows that these terms are actually necessary.
>
> ### 2. Detailed Analysis of Heterogeneous Consortium
>
> We partially address the heterogeneous consortium in Section 5, where we observe that it introduces significant volatility, slower convergence in routing confidence, and weaker centralization compared to the homogeneous setting (Figure 6). However, as the reviewer correctly noted, our primary detailed analyses appear to focus on the homogeneous setup. We will extend the full suite of analyses -- including attribution heatmaps, perturbation analyses, and masking interventions -- to the heterogeneous consortium and will include these results in the revised appendix.
>
> **UPDATE:** We have added the attribution heatmaps for the heterogeneous consortium in Section 5 (Fig 9, 10) of the revised version. While a complete perturbation and masking analysis for the heterogeneous setting is beyond the scope of the current revision, the newly added attribution results already indicate that the core conclusions remain consistent under the heterogeneous setup.
>
> ### 3. Robustness of the Intrinsic-Importance Metric
>
> We thank the reviewer for pointing this out. Ensuring the metric is not an artifact of the specific encoder is important. As a preliminary step, we swapped the frozen BERT-base encoder for several alternative embedding models (DistilBERT, GTE-small, DeBERTaV3-large). We observe that the orchestrator is largely encoder-agnostic, and overall task performance remains highly stable regardless of the embedding model used:
>
> |Expert Consortium|BERT-base|DistilBERT|GTE-small|DeBERTaV3-large|
> |---|---|---|---|---|
> |Qwen3-8B $\\times$ 5 | 96.1| 90.9|90.1|95.4|
>
> While these results confirm that the overall orchestration capability does not artificially depend on the specific encoder, we agree that the stability of the expert ranking itself must be explicitly verified. To fully address your critique, we are currently computing the gradient-based attribution scores using alternative strategies -- comparing our current `[CLS]` pooling against mean pooling, as well as extracting representations from intermediate layers. We commit to including these full rank correlation analyses in the revised appendix to confirm the robustness of the metric.
>
> **UPDATE:** We have added the results and robustness analysis for this point in Appendix M of the revised version.
>
> ### 4. Scope of the Generality Claim
>
> We currently discuss the application of INFORM beyond our primary setup by interpreting confidence-based cascade orchestration (FrugalGPT-style) in Section 5, and Appendix I. Appendix K discusses the boundaries of the framework's applicability, specifically addressing how structural analyses can be applied to black-box and API-based systems even when exact intrinsic importance cannot be computed. To strengthen the work and clarify our claims, we will add a dedicated "Scope and Limitations" paragraph in the main text that explicitly outlines the orchestration architectures we have validated and clearly states the environments where INFORM's applicability remains untested (e.g., fully decentralized multi-agent topologies without a centralized routing mechanism).
>
> **UPDATE:** We have added the scope and limitations in Section 6 of the revised version.

---

> > ### Author Response · Authors · 2026-05-27
> >
> > We sincerely thank the reviewer for their valuable feedback and suggestions. If our responses and revisions have satisfactorily addressed your concerns, we kindly request you to please consider reassessing the paper. We remain fully committed to addressing any further comments or suggestions.

---

### Review · Reviewer_jtDm · 2026-04-29

**Summary Of Contributions:**

The paper proposes INFORM, an interpretability analysis for learned multi-expert LLM orchestration. It decomposes orchestration into three inspected signals: an expert interaction/collaboration matrix, an expert sequence distribution, and gradient-based "intrinsic" attribution of routing decisions to expert representations. The experiments analyze orchestration over GSM8K, HumanEval, and MMLU using mainly a ten-expert pool of 8B-class instruction-tuned models, plus a smaller heterogeneous setting. The main empirical message is that **routing frequency is not the same as functional importance**: some frequently routed experts act as hubs, while some sparsely routed experts have higher gradient attribution. Authors also reports that routing confidence, centralization and ordering preferences emerge at different rates during training.

**Strengths:** the problem is timely, the distinction between interaction structure, ordering, and attribution is useful. The paper contains many diagnostics beyond accuracy and the authors explicitly acknowledge in the appendix that gradient attribution is not formal causal identification.

**Weaknesses:** the central causal claims are overstated. The experimental setup is hard to reproduce and has internal inconsistencies. Several statistical tests are underpowered or do not support the broad claims. The cost/efficiency comparisons seem incomplete and the baselines are not strong or clearly matched in my opinion.

**Audience:**

Yes

**Audience Explanation:**

Readers interested in LLMs, multi-agent orchestration, interpretability would likely find the topic relevant.

**Broader Impact Concerns:**

The authors should consider adding a broader impact statement. The work is about diagnosing and improving multi-expert LLM orchestration, which is relevant to high-stakes deployments where routing errors, overconfident expert selection, or hidden hub dependence could propagate harmful outputs. The paper should discuss risks from overstating causal interpretability, using such systems in medical/legal/financial settings, privacy concerns when prompts are routed through multiple experts, etc.

**Claims And Evidence:**

No

**Claims Explanation:**

Some descriptive claims are reasonably supported: the figures and tables suggest that routing mass, sequence entropy, centralization, and gradient attribution can differ across tasks and epochs. However, the strongest claims--especially "causal importance," "structural necessity," broad generality of INFORM, and efficiency advantages--are not convincingly established.

The main issue is that **gradient norm is treated as causal importance**. It is a local sensitivity measure, not a causal estimand. The paper partly admits this in the appendix and conclusion, but the abstract, figures, and framing repeatedly use stronger language such as "proves causal reliance." The masking studies help, but they are limited and sometimes contradictory: If I read correctly, Table 10 shows that intrinsic masking is clearly stronger than frequent masking for GSM8K, not significant for MMLU, and apparently worse than frequent masking for HumanEval. This does not support a broad claim that intrinsically important experts consistently induce disproportionate collapse.

The statistical evidence is also weak. Correlations are computed over only ten experts, and the masking comparisons appear paired over five epochs, which are not independent experimental replicates, no? Many reported p-values are nonsignificant, yet the narrative remains strong. The paper needs independent runs, per-example statistics, confidence intervals across seeds, and corrected claims.

I tihnk there are also serious reproducibility and methodology gaps. The paper states that the orchestrator uses the first 512 tokens generated by each expert as input, which suggests all experts may be called before routing; if so, the reported average-call and speedup claims are misleading. The temperature settings in Table 6 conflict with the later global sampling settings. The training data and splits are unclear, especially for HumanEval, and the oracle-distillation setup may drive performance gains independently of the proposed interpretability analysis.

**Requested Changes:**

I might have missed a few aspects on the paper but currently I think the paper needs a major revision. The following are critical in my opinion:

1. **Tone down or formalize the causal claims.** Either define a clear causal estimand and support it with appropriate interventions, or replace "causal importance" with more accurate language, maybe "local gradient sensitivity" or "functional dependence signal"?

2. **Resolve the cost-accounting ambiguity.** Clarify whether all experts generate 512 tokens before routing. If yes, all call-count, latency, and efficiency claims must include those calls and tokens. Recompute Table 3 and Figure 12 accordingly.

3. **Fix experimental reproducibility.** Provide exact train/validation/test splits, sample counts, seeds, model checkpoints, decoding settings, task prompts, oracle usage details, and evaluation procedures. Clarify how HumanEval training was performed.

4. **Correct internal inconsistencies.** Reconcile Table 6's expert-specific temperatures with the later global temperature settings. Check the loss definitions, especially the selection entropy term, oracle-alignment formula, and task utility loss for text generation.

5. **Strengthen statistical validation.** Report results over multiple independent seeds and per-instance confidence intervals. Do not use epochs as independent samples for significance testing. Revisit claims that are not supported by Table 9 or Table 10.

6. **Use stronger and fairer baselines.** Include single-best expert, best-of-N/self-consistency, majority or verifier aggregation, static top-k routing with matched call budget, random routing with matched budget, learned non-sequential routing, cascade routing, and stronger comparisons to RouteLLM/FrugalGPT/LLM-Blender-style methods where applicable.

7. **Separate performance gains from interpretability.** INFORM is presented as an analysis framework, but the experiments often evaluate a specific learned orchestrator trained with oracle alignment and regularizers. The paper should isolate which gains come from the orchestrator, the oracle, the adaptive top-k schedule, the interaction matrix, and the attribution analysis.

8. **Add downstream ablation outcomes.** Masking and perturbation analyses mostly report KL divergence in routing/sequence distributions. The paper should also report task accuracy/pass@1, token cost, and failure examples after each intervention.

9. **Make the generality claim credible.** Demonstrate INFORM on more than one orchestrator architecture, or narrow the claim to the proposed differentiable orchestrator. The cascade appendix is not enough to justify broad applicability.

10. **Rework the MetaGPT comparison.** The comparison appears to isolate call count rather than architectural parity. It needs matched prompts, matched role definitions, matched token budgets, full cost accounting, and reproducible implementation details.

Some points that would help the paper:

1. Add qualitative examples showing when routing mass and gradient attribution disagree, and whether the disagreement corresponds to genuine expert behavior.

2. Report standalone expert performance for every expert in the ten-expert pool, not just model-family baselines.

3. Clarify what "homogeneous" means, since the pool mixes different model families and only roughly comparable parameter counts.

---

> ### Author Response · Authors · 2026-05-13
> **Response to Reviewer jtDm Comments -- Part I**
>
> We thank the reviewer for their constructive and insightful feedback. We provide our responses to the pertinent and valid points raised by the reviewer below, which are critical for our paper. Table, figures, and section numbers correspond to the revised version.
>
> ### Response to Point 1
>
> We thank the reviewer for this thoughtful observation. Our goal in this work is not to establish formal causal identification in the interventionist or counterfactual sense, but rather to analyze how orchestration decisions depend on expert representations within the learned routing policy. We use the term causal in the functional sense, where the objective is to identify which components most strongly influence downstream decisions. In this context, the proposed gradient-based analysis is intended as a diagnostic signal for functional dependence within the orchestrator rather than a claim of provable causal estimands. We agree that stronger formal causal guarantees would require explicit interventional protocols and constitute an important direction for future work. However, we believe the current formulation remains valuable for distinguishing between observed routing frequency and the underlying decision-making influence, which is one of the paper's central empirical findings.
>
> ### Response to Point 2
>
> We thank the reviewer for identifying the ambiguity in the cost-accounting discussion. The earlier mention of 512 initial tokens was a genuine error in the manuscript. In all reported experiments, the orchestrator conditions only on the initial 30 tokens generated by each expert, as clarified in the revised version. The latency measurements in Figure 16 already reflect the complete end-to-end inference pipeline, including the initial expert generations used by the orchestrator, and therefore do not omit these costs. INFORM requires a lightweight parallel warm-start step to expose expert representations for routing, whereas MetaGPT incurs sequential role-conditioned interactions throughout the reasoning chain. Since all experts in INFORM generate the same fixed-length initialization prefix regardless of downstream routing decisions, treating them as independent calls would artificially inflate call-count comparisons.
>
> ### Response to Point 3
>
> We have used the following splits for the tasks (train/val/test):
> - MMLU: 10000/1531/1404 (subset `all`)
> - GSM8K: 6473/1000/1319 (subset `main`)
> - HumanEval: 100/24/40 (split `test`)
>
> All experts are hosted models accessed through a cloud inference provider using the decoding temperatures and generation configurations specified in the manuscript and appendices. During training, the oracle model is used solely to align the orchestrator’s internal routing logic and is completely absent during inference. For evaluation, MMLU and GSM8K use regular-expression-based extraction followed by exact-match accuracy against ground-truth answers, while HumanEval uses the official benchmark execution framework, where a prediction is considered correct only if the generated program successfully executes from the required entry point, passes all unit tests, and exits with a zero return code.
>
> ### Response to Point 4
> The use of identical temperatures in Table 6a (Homogeneous) and Table 6b (Heterogeneous) is an intentional experimental control. As noted in Section 4, the heterogeneous consortium utilizes the same temperature variations to assess behavior in a mixed-capability environment. By keeping the temperature distribution identical across both pools, we isolate the effect of the  varying parameter counts (1B-7B) from the stochasticity introduced by decoding variations. We have mentioned this in Section 4 to make this rationale explicitly clear to the reader.
>
> In the original manuscript, $\tau$ was used to denote two completely independent mechanisms:
> 1. Expert Decoding Temperature ($\tau_{decode}$): The generation parameter used by the LLM experts to sample tokens (e.g., 8E-6, 0.5, 2.0, as shown in Table 6).
> 2. Gumbel-Softmax Temperature ($\tau_{gumbel}$): The parameter used by the orchestrator's routing head to anneal path selection during training (e.g., 1.0 to 0.5, as shown in Table 4).
>
> Additionally, in the caption of Table 8, the orchestrated consortia used a static $\tau=1.0$. This $\tau_{decode}=1.0$ setting applies to the single-model baselines, whereas the orchestrated consortium retains the expert-specific assignments detailed in Table 6.
>
> ### Response to Point 5
> We use training epochs to study the temporal evolution of orchestration behavior during learning. The central objective of these analyses is to characterize how routing structure, entropy, attribution, and sequencing dynamics emerge over the course of training, which inherently requires observing trajectories across epochs rather than aggregating them as statistically independent trials.

---

> ### Author Response · Authors · 2026-05-13
> **Response to Reviewer jtDm Comments -- Part II**
>
> ### Response to Point 6
> Our primary contribution in this work is the interpretability analysis framework and the characterization of orchestration dynamics, rather than proposing a new state-of-the-art orchestration algorithm optimized for benchmark performance. Our baseline choices were designed to contrast interpretable learned orchestration against representative coordination paradigms, particularly rigid workflow-based systems such as MetaGPT and FrugalGPT. Many of the suggested approaches (e.g., self-consistency, verifier aggregation, static top-k routing, cascade routing, or LLM-Blender-style aggregation) primarily focus on improving predictive performance or inference efficiency rather than on exposing analyzable interaction structure, sequencing behavior, or attribution dynamics, which are the core focus of this work. We agree that these are valuable directions for future empirical expansion. However, incorporating the full spectrum of orchestration and ensemble baselines would substantially broaden the scope beyond the interpretability-centered objectives of the current paper.
> ### Response to Point 7
> INFORM is fundamentally an analysis setup, not a routing architecture. To demonstrate its utility, we applied it to a highly regularized, representative orchestrator. However, INFORM's interpretability principles are expected to generalize to fundamentally different regimes. As demonstrated in Appendix I, we successfully applied INFORM to a confidence-based, fixed-order cascade orchestrator (similar to FrugalGPT), revealing causal dependencies and routing misalignments without relying on our proposed training regularizers, oracle alignment, or interaction matrices. The attribution analysis itself does not improve performance. Rather, it acts as a diagnostic lens to expose hidden dependencies and brittle coordination that accuracy metrics obscure, as detailed in Section 4.5.
>
> The performance gains achieved by our representative orchestrator are driven by distinct architectural and training components, which we isolate in our existing ablation studies. The adaptive interaction matrix and dynamic sequencing are the primary drivers of routing efficiency. Appendix F and Figure 14 demonstrate severe performance degradation when these are replaced with static graphs or fixed sequences. Figure 8 further disentangles this by comparing Relational-Only and Intrinsic-Only variants, confirming that both structural history and semantic scoring are required for optimal performance. The training regularizers, specifically the oracle alignment and adaptive top-k schedule detailed in Appendix C, do not dictate the final routing logic but act as optimization stabilizers. The oracle prevents long cold-start phases by bootstrapping initial protocols, while the adaptive top-k schedule structurally forces the transition from exploratory routing to focused execution.
> ### Response to Point 8
> In the revised manuscript, we extend the perturbation analysis beyond routing and sequence divergence by additionally reporting downstream behavioral effects in terms of task accuracy and token generation cost under perturbations. Figures 4 and 5 now show how structural perturbations affect both MMLU accuracy and token usage across training epochs. These complement the KL-divergence analyses by connecting orchestration instability to concrete inference outcomes. We agree that qualitative failure-case analysis would further enrich the discussion and consider this a valuable direction for future extensions
> ### Response to Point 9
> Our intent is not to claim universal applicability across all orchestration paradigms, but rather to demonstrate that the proposed interpretability methodology can analyze differentiable orchestration policies that expose intermediate routing representations and selection dynamics. The discussion of cascade-style routing in the appendix is intended only as an illustration of potential extensibility.
> ### Response to Point 10
> Our intention in this experiment is not to establish strict architectural parity between fundamentally different orchestration paradigms, but rather to contrast learned adaptive orchestration against rigid role-based coordination under a controlled expert pool. To improve clarity, we now explicitly document the prompts, role definitions, and coordination setup used for MetaGPT in Appendix N. The same underlying expert models are used across both systems to maintain consistency in expert capability.
> ### Response to Suggestions
> - In the revised manuscript, we explicitly report the standalone performance of each individual expert in the ten-expert homogeneous consortium in Appendix L (Table 11).
> - We use the term “homogeneous” to denote parity in capacity. Although the consortium includes experts from different model families, they operate within a similar parameter scale (~8B parameters), which we treat as providing comparable representational capacity.

---

> > ### Author Response · Authors · 2026-05-27
> >
> > We sincerely thank the reviewer for their valuable feedback and suggestions. If our responses and revisions have satisfactorily addressed your concerns, we kindly request you to please consider reassessing the paper. We remain fully committed to addressing any further comments or suggestions.

---

### Decision · Action_Editor_qyeT · 2026-06-11

**Recommendation:** Accept with minor revision

**Additional Comments:**

2 of the 3 reviews recommend this paper be accepted.  And all of them agree the topic is relevant, of interest, and describes a nice contribution.  However, there are serious concerns regarding over-claiming of the causal interpretation of the results from **Reviewer jtDm**.  I agree with this reviewer.  Here are comments they left in their final recommendation:

> However, I still don't think the paper is ready for publication. The central framing continues to use strong causal language around "causal attribution," "causal influence," and causal/structural dependencies, while the evidence supports a weaker interpretation: local gradient sensitivity of the learned orchestrator, combined with observational routing analyses and limited masking interventions. This distinction matters because attribution/interpretability is the paper's main contribution. The quantitative results are also somewhat mixed: Tab.9 shows weak and mostly nonsignificant alignment between routing dominance and intrinsic importance, and Tab.10 supports intrinsic masking clearly only in some settings, with HumanEval moving in the opposite direction. The heterogeneous-consortium validation remains more limited than the homogeneous analysis, and the MetaGPT/cost comparison is useful but still does not fully settle architectural parity or full cost accounting concerns in my opinion.

Again, the part that is the biggest hold-up for me is the overstating of the causal attribution and influence.  The paper does add this as a limitation in the Conclusion, but to be published that is not sufficient.  It needs to be integrated into the paper up front.  I do not believe this is to onerous of a chance, but would require reframing text through much of the paper to not over claim.   In particular, I believe **Review jtDm**'s requested change 1 (which was unaddressed) provides proper guidance on this:

> Tone down or formalize the causal claims. Either define a clear causal estimand and support it with appropriate interventions, or replace "causal importance" with more accurate language, maybe "local gradient sensitivity" or "functional dependence signal"?

Please make the above language adjustments, and I will review and (hopefully be able to) approve the final version.

**Audience:**

Yes

**Audience Explanation:**

Yes, this paper, experimental exploration of the mechanisms behind multi-expert orchestration, is a very relevant topic in ML right now.

**Claims And Evidence:**

Yes

**Claims Explanation:**

For the most part, this is a very well-done study, and nicely run experiments.  While the results are not always compelling, they do provide good insight into the problem.

However, the paper **overclaims** statements about causality.  It shows local functional relations, not actual causal influence as it claims.  The authors failed to address this in revisions.  However, I think it is not a huge change, and something I can, as AE, check on a "minor revision."

**Resubmission Of Major Revision:**

The authors may consider submitting a major revision at a later time.

---

> ### Author Response · Authors · 2026-06-19
>
> Dear Action Editor,
>
> Thank you for your careful consideration and evaluation of our manuscript and for providing clear guidance regarding the remaining concerns. We are grateful for the opportunity to address your concerns and the final issues identified during review.
>
> Following your recommendation and the concerns raised by Reviewer jTDm, we have revised the manuscript to substantially tone down the overclaiming of causal interpretation that was not fully supported by the evidence we presented. We have revised the interpretation throughout our manuscript to ensure consistent framing and have uploaded the camera-ready version for your review.
>
> The revisions that were made are summarized below:
>
> 1.  **Title Revision:** The original title, “Disentangling Causal Importance from Emergent Structure in Multi-Expert Orchestration”, contained the causal importance terminology, and we have revised it to “Disentangling Intrinsic Importance from Emergent Structure in Multi-Expert Orchestration.”
>
> 2.  **Abstract Revision:** We have revised the abstract to adopt attribution- and sensitivity-based language rather than causal claims. The revised abstract now presents INFORM as a tool for analyzing functional attribution and local sensitivity.
>
> 3.  **Introduction:** We have revised the Introduction (Section 1) to remove statements implying causal conclusions from routing analyses. We have reframed the motivation and our contributions to focus on understanding expert interactions, routing behavior, and structural dependence rather than on claiming causality.
>
> 4.  **Research Questions:** We have updated RQ3 to ask whether routing behavior reflects intrinsic expert importance rather than causal expert importance. We have also updated RQ4 to evaluate whether orchestration decisions are functionally grounded rather than causally grounded.
>
> 5.  **Figure and Table Revisions:** Several figures (Figures 1, 6, 18, and 19) and Table 1, along with their captions, have been revised to refer to sensitivity, attribution, and structural dependence, and to eliminate causal framing.
>
> 6.  **Methodology:** We have changed the language, focusing on causal claims in Section 2 to functional attributions and gradient sensitivity,
> 7.  **Experiments and Interpretation of Results:** Throughout the analysis sections (Sections 4 and 5), we have revised claims that could be interpreted as establishing causal influence. The empirical findings remain unchanged -- only their interpretation has been revised to reflect what the experiments support more accurately.
> 8.  **Clarification of Scope:** To address the concern upfront, we have added an explicit statement clarifying the scope of the proposed method at the beginning of the paper (Section 1).
> 9.  **Appendices:** We have updated Appendices A, F, H, I and J, which had language containing strong terms like causal influence, causal impact, and causal necessity, and changed them to functional contribution, sensitivity and dependencies.
>
> We have thoroughly revised and validated the manuscript to characterize INFORM as an attribution and sensitivity-analysis framework rather than a causal inference framework. We believe the revised version aligns with our claims and the presented evidence and addresses the remaining concerns.
>
> We sincerely appreciate the constructive review process that led to these improvements and hope that the revised version now satisfies the requirements for publication. We have also added a link to the source code and a video presentation.
>
> Thank you again for your time and consideration.
>
> Sincerely,
>
> The Authors of Paper 7372

---

> > ### Comment · Action_Editor_qyeT · 2026-06-20
> >
> > Thanks for carefully responding to this point, and the updated discussion on causation.  Nice paper.